# ITK signalling via the Ras/IRF4 pathway regulates the development and function of Tr1 cells

Weishan Huang[1,2], Sabrina Solouki[2], Nicholas Koylass[2], Song-Guo Zheng[1,3] & Avery August[2]

Type 1 regulatory T (Tr1) cells differentiate in response to signals engaging the T cell receptor (TCR), express high levels of the immunosuppressive cytokine IL-10, but not Foxp3, and can suppress inflammation and promote immune tolerance. Here we show that ITK, an important modulator of TCR signalling, is required for the TCR-induced development of Tr1 cells in various organs, and in the mucosal system during parasitic and viral infections. ITK kinase activity is required for mouse and human Tr1 cell differentiation. Tr1 cell development and suppressive function of *Itk* deficient cells can be restored by the expression of the transcription factor interferon regulatory factor 4 (IRF4). Downstream of ITK, Ras activity is responsible for Tr1 cell induction, as expression of constitutively active HRas rescues IRF4 expression and Tr1 cell differentiation in $Itk^{-/-}$ cells. We conclude that TCR/ITK signalling through the Ras/IRF4 pathway is required for functional development of Tr1 cells.

[1] Center for Clinical Immunology, The Third Affiliated Hospital of Sun Yat-sen University, Guangzhou, Guangdong 510630, China. [2] Department of Microbiology and Immunology, Cornell University, Ithaca, New York 14853, USA. [3] Department of Medicine, Milton S. Hershey Medical Center, Pennsylvania State University, Hershey, Pennsylvania 17033, USA. Correspondence and requests for materials should be addressed to W.H. (email: weishan.huang@cornell.edu) or to A.A. (email: averyaugust@cornell.edu).

regulatory (Treg) cells promote immune tolerance and suppress inflammation[1,2]. Unlike Treg cells that stably express the transcription factor Foxp3, type 1 regulatory T (Tr1) cells have no or transient expression of Foxp3; however, they produce high levels of IL-10 and can suppress effector cell responses in an IL-10 dependent manner[1,3], via CTLA-4 and PD-1 interactions, or by directly killing pro-inflammatory cells with granzymes[2,4]. In mice and in humans, induction of antigenic tolerance during haematopoietic stem cell transplantation and specific-antigen immunotherapy are positively correlated with the abundance of Tr1 cells[5,6], and Tr1 cells can prevent allergic asthma induced by the house dust mite peptidase 1 variant Derp 1 in murine models[7], and prevent the development of bacterial-induced atopic dermatitis[8]. Thus, like Foxp3[+] Treg cells, Foxp3[−]IL-10[+] Tr1 cells have therapeutic potential for inflammatory diseases. Although much is known about the development and function of Treg cells, substantially less is known about Tr1 cells. A better understanding of the development and function of Tr1 cells should provide a wider array of therapeutic options for inflammatory diseases.

IL-2 inducible T cell kinase (ITK) is a Tec family non-receptor tyrosine kinase expressed by T cells, and has a pivotal role downstream of the T cell receptor (TCR); the loss of ITK function leads to attenuated TCR signalling and alters the T cell subset differentiation and function[9]. Naive CD4[+] T cells can differentiate into Tr1 cells upon TCR engagement in the presence of IL-27, and although Tr1 cells can express IFN-γ, production of IFN-γ or T-bet are not required for Tr1 cell development[10]. Alternatively, Tr1 cells can result from Th17 trans-differentiation during the resolution of inflammation[11]. These findings suggest that Tr1 cell differentiation may share some pathways of regulation with Th1 and Th17 cell development. In mice with ITK deficiency, naive CD4[+] T cells have defects in the differentiation of Th17 cells[12], and enhanced Th1 differentiation with impaired Th2 and Th9 programming that leads to attenuated allergic asthma[13–15], and have enhanced differentiation of Foxp3[+] Treg cells[16,17]. Whether ITK also has a function in modulating the development and/or function of IL-10-producing Tr1 cells, is unexplored.

Beyond the finding that the cytokine IL-27 and the transcription factors interferon regulatory factor 4 (IRF4), avian musculoaponeurotic fibrosarcoma (cMAF) and aryl hydrocarbon receptor (AHR) are important for Tr1 cell differentiation, we have limited knowledge of the signalling pathways that regulate the development and, importantly, function of Tr1 cells. Here we show that, in the absence of ITK, TCR engagement does not induce optimal differentiation of Tr1 cells in multiple organs and during parasitic or viral infection. The expression and activity of ITK are crucial for Tr1 cell fate programming in both mouse and human, and for Tr1 cell function to suppress effector cell expansion. ITK deficiency impairs IRF4 expression in both mouse and human Tr1 cell development, and restoring IRF4 expression rescues Tr1 cell fate programming and suppressive function in Itk deficient cells. The RAS/MAPK signalling axis is indispensable for Tr1 cell development, and constitutively active RAS signalling completely rescues induction of IL-10 and IRF4 during Tr1 cell differentiation of Itk deficient cells. Our findings identify ITK as a crucial component that bridges extracellular signals, RAS signalling and IRF4 expression during Tr1 cell fate programming, and suggest that ITK signalling components are potential targets for modulating Tr1 cell development and function for clinical benefit.

## Results

**ITK is required for Tr1 cell development in vivo.** We and others have previously shown ITK as a negative regulator of Foxp3[+]

Treg cell development[16,17], but its role in CD4[+]Foxp3[−]IL-10[+] Tr1 cell development is unclear. To determine the role of ITK in Tr1 cell differentiation in vivo, we injected WT and Itk[−/−] IL-10[GFP]/Foxp3[RFP] dual reporter mice with an anti-CD3ε antibody that has been shown to stimulate pronounced Tr1 cell development through TCR activation in vivo[3]. We observed a significant defect in IL-10[+]Foxp3[−] LAG3[+]CD49b[+] Tr1 cell differentiation systemically in blood, spleen, lung, small intestine and fat, in mice lacking the expression of ITK (Fig. 1), suggesting that ITK is required for Tr1 cell development in vivo. IL-10 production can be significantly elevated in the pulmonary mucosa during the late stages of Nippostrongylus brasiliensis[18] and influenza A[19] infection to prevent tissue damage. To determine whether ITK is required for Tr1 cell development during these infections, we challenged mice with N. brasiliensis larvae or influenza A (WSN) virus, and found that ITK is required for Tr1 cell differentiation during parasitic (Fig. 2a,b) and viral (Fig. 2c,d) infections. Our data support a requisite role for ITK in Tr1 cell development in vivo.

**ITK is required for Tr1 cell fate programming in vitro.** To determine whether ITK regulates Tr1 cell development in vitro, we isolated naive splenic CD4[+] T cells from WT and Itk[−/−] mice carrying IL-10[GFP]/Foxp3[RFP] dual reporters, and cultured them under Tr1-polarizing condition in vitro. We found that in the absence of ITK, differentiation of IL-10[+] Foxp3[−] Tr1 cells is severely impaired, which persisted along the time course of our observation (Fig. 3). A similar defect in Tr1 cell differentiation was also observed using naive T cells isolated from the thymus (Supplementary Fig. 1). ITK is not required for early activation of CD4[+] T cells under Tr1-differentiation condition, as Itk[−/−] CD4[+] T cells effectively up-regulated early activation markers CD25 and CD69 within 48 h post stimulation (Fig. 3c), and all cells proliferated within 72 h post stimulation (Fig. 3d). However, in the absence of ITK, cells that proliferated failed in the induction of IL-10 production (Fig. 3d, bottom panel), and are impaired in expression of the Tr1-related markers LAG3, CD49b, ICOS and PD-1 (Fig. 3e,f). These data are suggestive of a requisite role for ITK in Tr1 cell fate programming that is not explained by a lack of T cell activation.

**ITK kinase activity is required for Tr1 cell differentiation.** We have recently shown that the absence of ITK in T cells can have different functional outcomes compared with inhibition of its kinase activity[20]. Furthermore, the ITK inhibitors that have been developed to date all exhibit varying degrees of cross reactivity with other tyrosine kinases (see review ref. 21). To get around these caveats, we developed a unique transgenic mouse system that expresses an altered form of ITK, referred to as allele sensitive ITK (or ITK$_{as}$) that allows us to inhibit the kinase activity of ITK using the small molecule 3MBPP1 (refs 16,20). To definitively and unambiguously determine the consequence of inhibiting the kinase activity of ITK during Tr1 cell development, we generated Itk$_{as}$ IL-10[GFP]/Foxp3[RFP] dual reporter mice. Using naive CD4[+] cells from these mice, we found that specifically inhibiting the kinase activity of ITK resulted in similar defect in Tr1 cell differentiation to that seen in the absence of ITK expression (Fig. 4a). They also exhibited severely impaired expression of classically defined surface markers of Tr1 cells, LAG3 and CD49b[1] (Fig. 4b). Such defective Tr1 cell differentiation is unlikely because of the lack of TCR-mediated cell survival and proliferation but rather impaired Tr1 cell fate programming, because CD4[+] T cells survival and proliferation remained similar in the absence of ITK expression or its kinase activity, compared with the case for WT T cells (Fig. 4b). We also observed that the absence of ITK or its kinase activity resulted in a

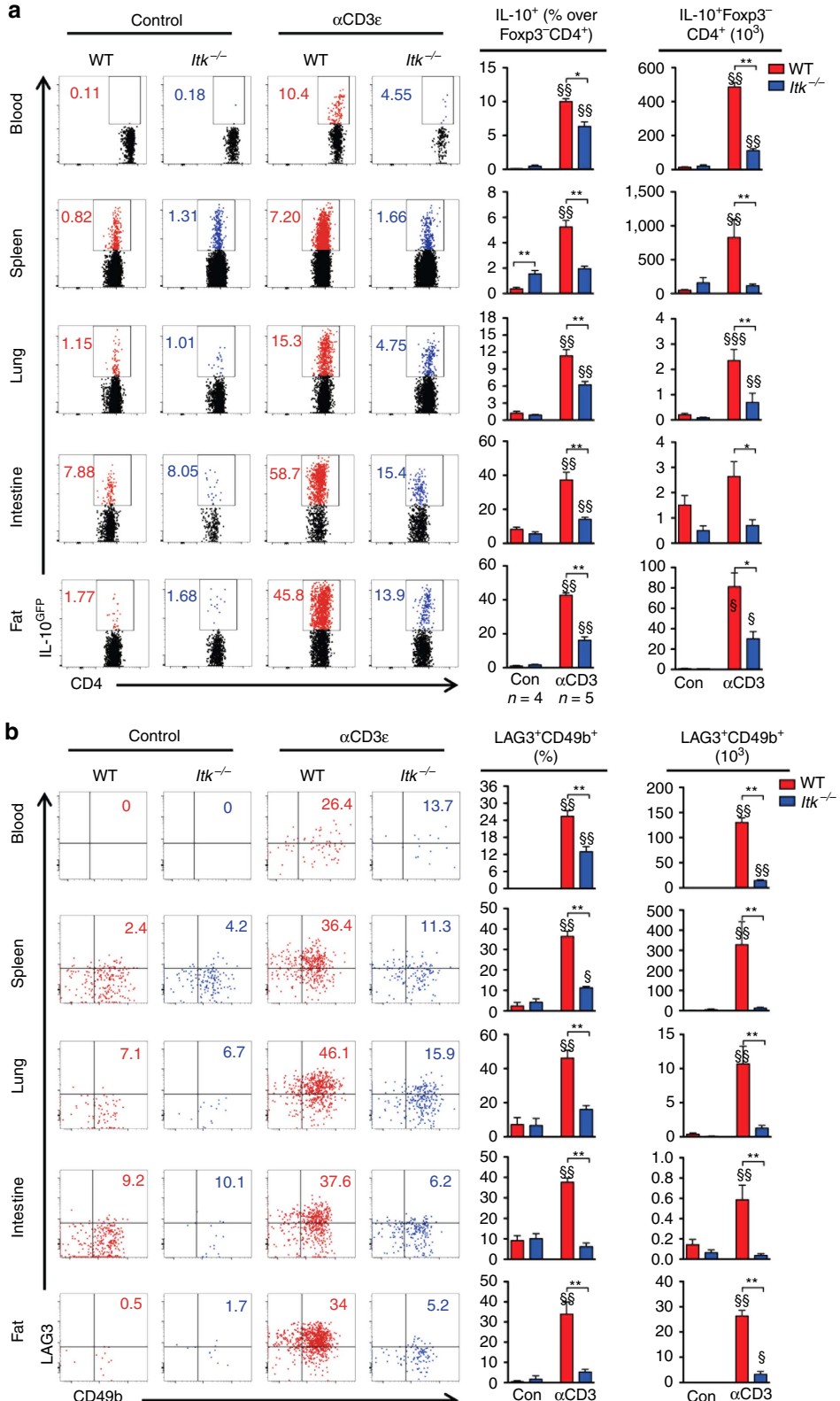

**Figure 1 | ITK is required for Tr1 cell development *in vivo*.** WT and *Itk*[−/−] IL-10[GFP]/Foxp3[RFP] dual reporter mice were treated with αCD3ε antibody or control, and live cells from the indicated organs were analysed. (**a**) Representative FACS plots showing IL-10[GFP] expression by Foxp3[RFP]− CD4[+] T cells and summary of IL-10[+] cell percentage over Foxp3[−] CD4[+] T cells and number in blood (number per ml), spleen, lung, small intestine and fat (number per gram). (**b**) Representative FACS plots showing the LAG3/CD49b expression by IL-10[GFP]+ Foxp3[RFP]− CD4[+] T cells, summary of LAG3[+]CD49b[+] cell percentage over IL-10[+]Foxp3[−]CD4[+] T cells and number of LAG3[+]CD49b[+] IL-10[+]Foxp3[−] Tr1 cells from samples shown in (**a**). [§]$P \leq 0.05$, [§§]$P \leq 0.01$, [§§§]$P \leq 0.001$, compared with the levels in PBS-treated group; [*]$P \leq 0.05$, [**]$P \leq 0.01$, [***]$P \leq 0.001$, comparing groups connected, by non-parametric Mann–Whitney test. Data were pooled from three experiments, '*n*' indicates number of replicates in each group/point. Data presented as mean ± s.e.m.

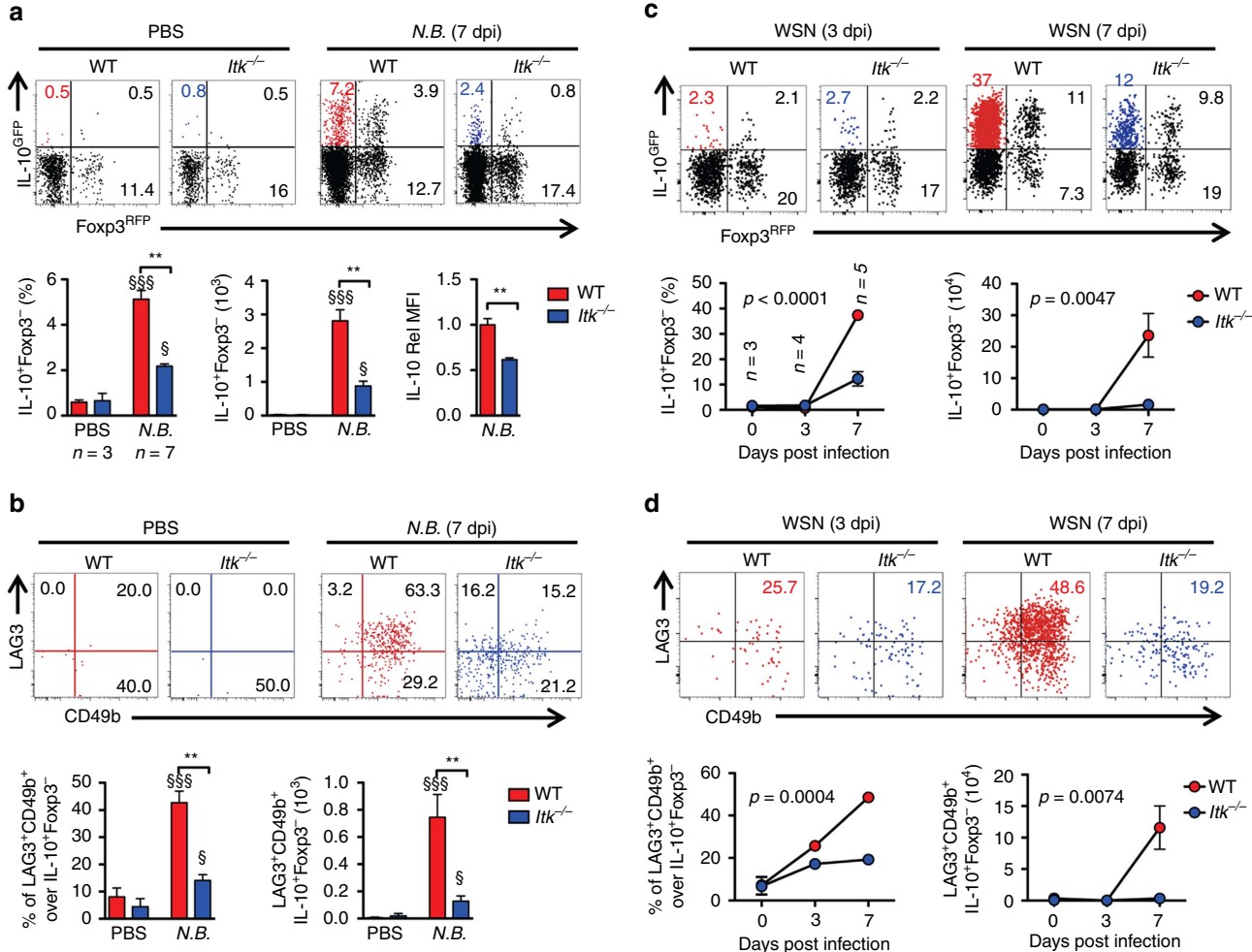

**Figure 2 | ITK is required for parasitic/viral infection-induced Tr1 cell development *in vivo*. (a,b)** ITK is required for Tr1 cell development during parasitic infection: WT and $Itk^{-/-}$ mice were infected with 500 L3 *N. brasiliensis* (*N.B.*) and lungs analyzed 7 days post infection (d.p.i.). **(a)** Representative FACS plots showing IL-10$^{GFP}$ and Foxp3$^{RFP}$ expression by pulmonary CD4$^+$ T cells and summary of IL-10$^+$Foxp3$^-$ T cell percentage, number and IL-10 expression levels (WT average level set as 1). **(b)** Representative FACS plots showing the LAG3 and CD49b expression by IL-10$^{GFP+}$ Foxp3$^{RFP-}$ CD4$^+$ T cells, summary of LAG3$^+$CD49b$^+$ IL-10$^+$Foxp3$^-$CD4$^+$ Tr1 cell percentage and number from samples shown in **a**. $^{\S}P \le 0.05$, $^{\S\S}P \le 0.01$, $^{\S\S\S}P \le 0.001$, compared with PBS-treated group; $*P \le 0.05$, $**P \le 0.01$, $***P \le 0.001$, comparing groups connected, by non-parametric Mann-Whitney test. **(c,d)** ITK is required for Tr1 cell development during viral infection: WT and $Itk^{-/-}$ mice were infected with 10$^4$ PFU Influenza A/WSN/1933 (WSN) and lungs analyzed 3 and 7 d.p.i. **(c)** Representative FACS plots showing IL-10$^{GFP}$ and Foxp3$^{RFP}$ expression by pulmonary CD4$^+$ T cells and summary of IL-10$^+$Foxp3$^-$ T cell percentage, number and IL-10 expression levels (WT average level set as 1). **(d)** Representative FACS plots showing the LAG3/CD49b expression by IL-10$^{GFP+}$ Foxp3$^{RFP-}$ CD4$^+$ T cells, summary of LAG3$^+$CD49b$^+$ IL-10$^+$Foxp3$^-$CD4$^+$ Tr1 cell percentage and number from samples shown in **c**. $P$ values on plots were calculated by two-way ANOVA. Data were pooled from three experiments, '$n$' indicates number of replicates in each group/point. Data presented as mean ± s.e.m.

significant shift in the production of IFN-γ instead of IL-10 under Tr1 polarizing conditions, although this was not induced by TCR re-activation (Supplementary Fig. 2), suggestive of altered commitment to Tr1 cell fate decision during primary stimulation. In human cells, inhibiting ITK kinase activity using the selective ITK inhibitor BMS-509744 (ref. 22) or the broad Tec kinase inhibitor CNX584 (ref. 23) revealed that the kinase activity of ITK is required for IL-10 production and human Tr1 cell differentiation as well (Fig. 4c,d), suggesting a conserved requisite role for ITK activity in Tr1 cell fate programming in human and mouse.

**ITK kinase activity required for Tr1 cell suppressive function.** Tr1 cells can suppress effector T cell function in an IL-10-dependent manner[1,3], but we have little knowledge of how this

function is regulated. Because of the lack of methods to isolate viable IL-10 producers and specially target them in a co-culture system in the past, the main efforts in the field have been devoted to Tr1 cell development as an end point. In our experimental platform, the use of ITK$_{as}$ and its specific inhibitor 3MBPP1 would allow us to specifically target ITK kinase activity in $Itk_{as}$ Tr1 cells when they are co-cultured with responding effector cells. Using FACS sorted IL-10-producing $Itk_{as}$ Tr1 cells, we were able to examine the role of ITK activity in Tr1 cell suppressive function. We found that $Itk_{as}$ Tr1 cells are able to suppress effector T cell expansion as efficiently as WT Tr1 cells, and importantly, that inhibiting the kinase activity of ITK in the $Itk_{as}$ Tr1 cells by 3MBPP1 significantly diminished their ability to suppress (Fig. 5a,b), accompanied by a reduction in IL-10 production (Fig. 5c). These data indicate that ITK activity is required for functional suppression by Tr1 cells.

**ITK induced IRF4 is critical for Tr1 cell development and function.** Unlike Foxp3 in Foxp3-expressing Treg cells, the transcription factors regulating Tr1 cells are less clear. AHR and

cMAF have been shown to promote the IL-27-induced Tr1 differentiation[24], but although AHR expression was reduced during Tr1 cell development in the absence of ITK function (Fig. 6a), an

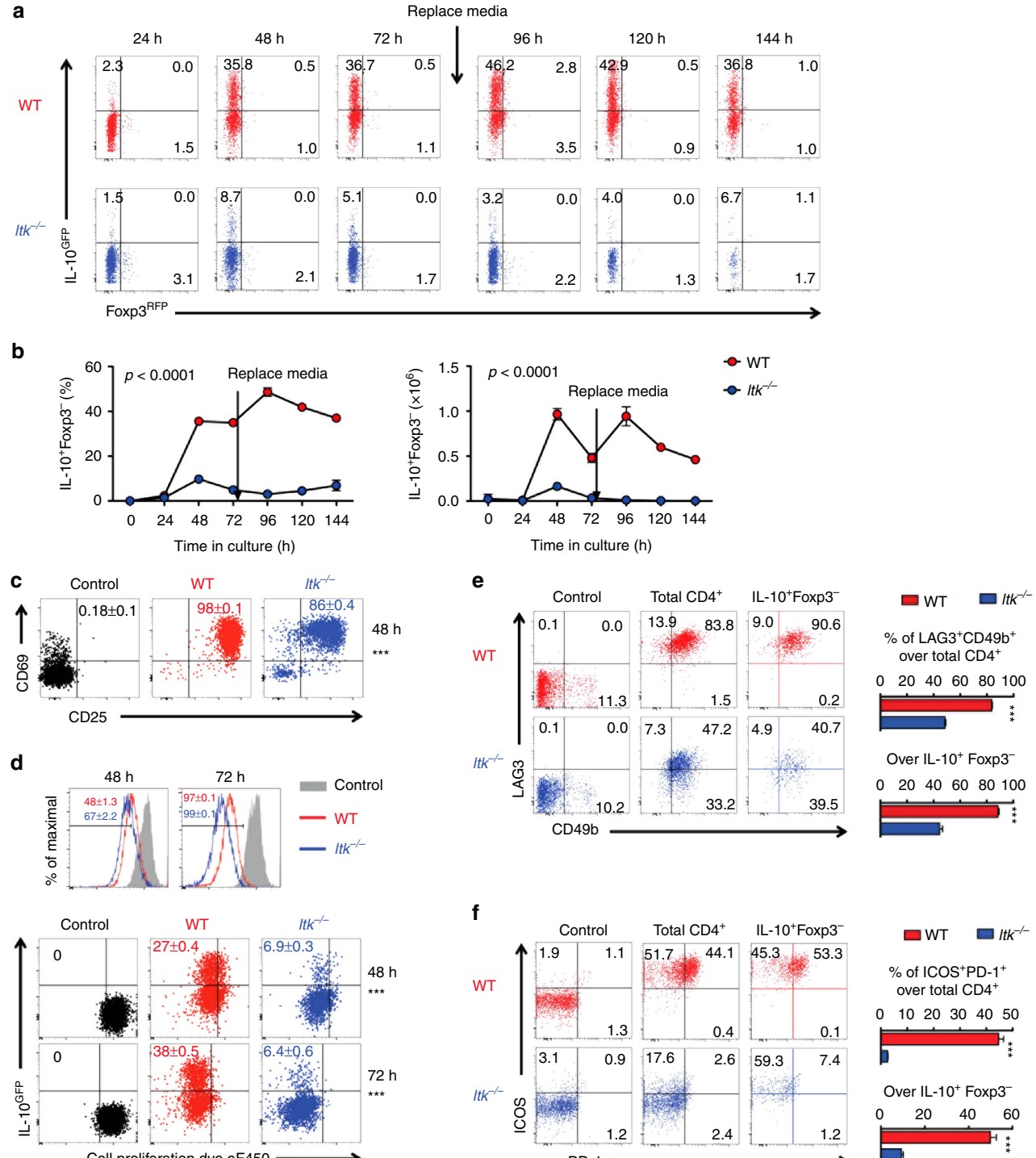

**Figure 3 | ITK is required for Tr1 cell differentiation *in vitro*.** All experiments were performed with cells carrying the IL-10$^{GFP}$/Foxp3$^{RFP}$ dual reporter system for live cell analysis. Naive WT and $Itk^{-/-}$ CD4$^+$ T cells were cultured under Tr1 polarizing condition. (**a**) Representative FACS plots showing the IL-10$^{GFP}$ and Foxp3$^{RFP}$ expression by naive CD4$^+$ T cells cultured under Tr1 differentiation conditions for the indicated time. (**b**) Summary of IL-10$^+$Foxp3$^-$ cell percentage and number (number per ml, initial naive CD4$^+$ T cell density: 0.5 × 10$^6$ per ml). $p$ values calculated by two-way ANOVA. (**c**) Representative plots of CD25 and CD69 expression by naive CD4$^+$ T cells cultured under Tr1 cell-differentiation conditions for 48 h. (**d**) Representative FACS plots showing the dilution of cell proliferation dye eF450 and expression of IL-10 by CD4$^+$ T cells cultured under Tr1 differentiation conditions for 48 or 72 h. Mean ± s.e.m. of six replicates were indicated on the flow cytometric plots. Representative plots and summary of percentages of double positive of (**e**) LAG3/CD49b and (**f**) ICOS/PD-1 expression by the total and IL-10$^+$Foxp3$^-$ CD4$^+$ T cells 72 h post culture. Naive CD4$^+$ T cells were used as controls in **c**–**f**. $n = 6$. Data represent results of more than three experiments. ***$P \leq 0.001$, by non-parametric Mann–Whitney test. Data presented as mean ± s.e.m.

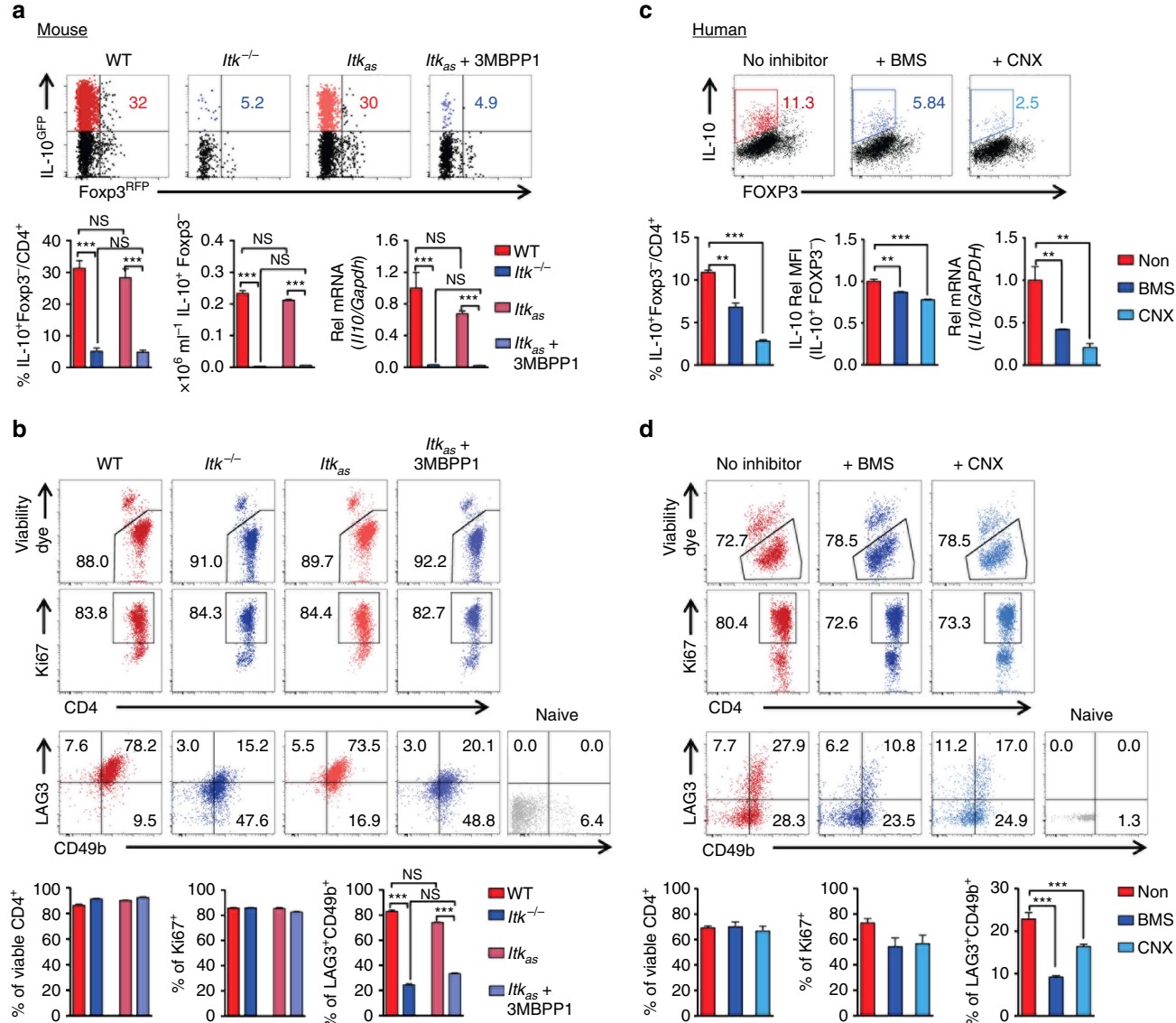

**Figure 4 | ITK is required for Tr1 cell differentiation in a kinase dependent manner.** (**a,b**) ITK kinase activity is required for Tr1 cell differentiation in mouse: WT, $Itk^{-/-}$ and $Itk_{as}$ naive CD4$^+$ T cells (carrying IL-10$^{GFP}$/Foxp3$^{RFP}$ reporters) were cultured under Tr1 polarizing conditions. (**a**) Representative FACS plots showing IL-10 and Foxp3 expression and summary of IL-10$^+$Foxp3$^-$ cell percentage, density (initial density: $0.5 \times 10^6$ per ml) and IL-10 relative (Rel) mRNA levels (normalized to Gapdh first, then WT average level set as 1). (**b**) Representative plots of CD4$^+$ T cell viability (first panel), expression of Ki67 (second panel) and LAG3/CD49b (third panel) in viable CD4$^+$ T cells; along with summary of percentage of viable CD4$^+$, Ki67$^+$ CD4$^+$ and LAG3$^+$CD49b$^+$ CD4$^+$ T cells. $n = 6$. Data represent results of more than five experiments. (**c,d**) ITK kinase activity is required for human Tr1 cell differentiation: naive CD4$^+$ T cells isolated from human peripheral blood mononuclear cells were cultured under Tr1 polarizing conditions; cells were stimulated and subjected to intracellular staining. (**c**) Representative FACS plots showing IL-10 and FOXP3 expression and summary of IL-10$^+$FOXP3$^-$ cell percentage, IL-10 Rel MFI levels (Non-treated group average level set as 1) and IL-10 Rel mRNA levels (normalized to GAPDH first, then WT average level set as 1). (**d**) Representative plots of CD4$^+$ T cell viability (first panel), expression of Ki67 (second panel) and LAG3/CD49b (third panel) in viable CD4$^+$ T cells; along with summary of percentage of viable CD4$^+$, Ki67$^+$ CD4$^+$ and LAG3$^+$CD49b$^+$ CD4$^+$ T cells. $n = 4$. Data represent results of two experiments. $^*P \leq 0.05$, $^{**}P \leq 0.01$, $^{***}P \leq 0.001$, by non-parametric Mann–Whitney test. Data presented as mean $\pm$ s.e.m.

AHR agonist (or antagonist) was not able to rescue Tr1 differentiation in the absence of ITK (Fig. 6b). cMAF in the other hand, showed an interesting discrepancy in mouse versus human CD4$^+$ T cells during Tr1 cell differentiation (Fig. 6c,d), which may be due to the difference in mouse and human Tr1 polarizing conditions used. Nevertheless, given that ITK kinase activity is required for both human and mouse Tr1 cell development (Fig. 4), it is unlikely that defective cMAF expression down-stream of ITK is responsible for the defect of Tr1 cell development specifically in the mouse. We did however, observe a significant reduction in IRF4 expression when ITK is

absent or inhibited in both mouse and human during Tr1 cell development (Fig. 6e,f).

In CD4$^+$ T cells, IRF4 up-regulates IL-4 and IL-10 expression under Th2 polarizing condition[25]; and IRF4 expression is reduced during activation of $Itk^{-/-}$ CD8$^+$ T cells[26] and Th9 cell differentiation[15]. However, it is unclear whether IRF4 expression alone is responsible for the Tr1 deficient phenotype we observe in the absence of ITK. Thus to determine whether IRF4 expression is down-stream of ITK signalling during Tr1 cell differentiation, we transduced naive CD4$^+$ T cells with retroviral particles delivering IRF4-YFP (or YFP alone as control). The transduction restored

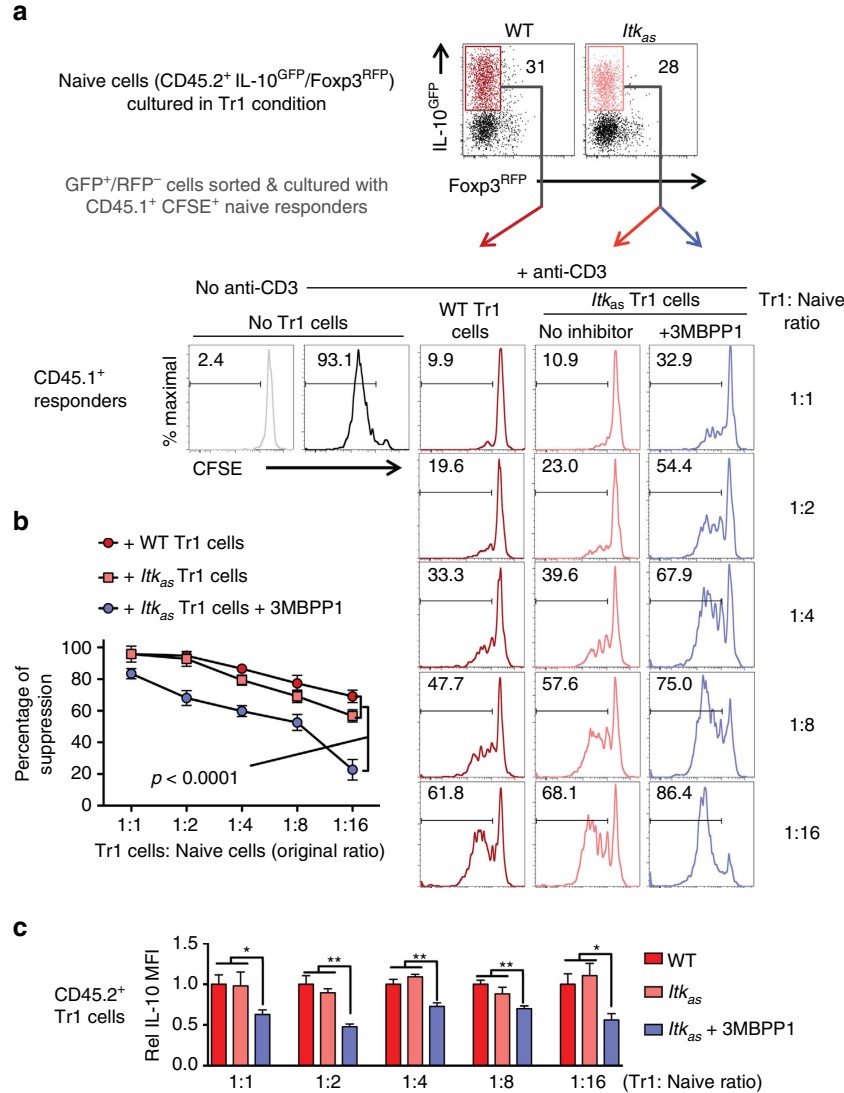

**Figure 5 | The kinase activity of ITK is required for Tr1 cell suppressive function.** Differentiated and sort purified WT and $Itk_{as}$ Tr1 cells (CD45.2[+], carrying IL-10[GFP]/Foxp3[RFP] reporters) were co-cultured with CFSE stained naive CD4[+] T cell responders (CD45.1[+]). 3MBPP1 was added to specifically inhibit $Itk_{as}$ activity. (**a**) Representative FACS plots of WT and $Itk_{as}$ Tr1 cells and CFSE dilution of the CD45.1[+] responders with or without stimulation in the presence or absence of ITK-inhibited Tr1 cells or control cells (responder cell density was fixed and Tr1 cells were added at the indicated ratios). (**b**) Summary of percentage of suppression of the responder proliferation by the ITK-inhibited Tr1 cells or control cells. $P$ value calculated by two-way ANOVA. (**c**) Relative (Rel) IL-10 MFI in CD45.2[+] Tr1 cells in the co-culture system. WT average level was set as 1. *$P \leq 0.05$, **$P \leq 0.01$, ***$P \leq 0.001$, by non-parametric Mann–Whitney test. $n = 6$. Data represents results of three experiments. Data presented as mean ± s.e.m.

IRF4 expression in CD4[+] T cells cultured under Tr1-polarizing condition (Supplementary Fig. 3a,b) and we found that the re-expression of IRF4 in $Itk$ deficient cells rescued Tr1 cell development (Fig. 7a,b), suggesting that ITK signalling-driven IRF4 expression is essential for Tr1 cell differentiation. To further determine whether re-expression of IRF4 rescued the function of Tr1 cells as well, we isolated cells that were transduced with IRF4-RV[YFP+], along with controls, and performed the *in vitro* Tr1 suppression assay. Our results showed that $Itk^{-/-}$ Tr1 cells that differentiated upon re-expression of IRF4 were fully functional in suppressing responding T cell proliferation (Fig. 7c). Thus, downstream of ITK, IRF4 is required for Tr1 cell functional developmental.

**ITK induced Blimp-1 does not drive Tr1 cell differentiation.** We have also noted that when IRF4 was re-expressed in $Itk^{-/-}$ T cells, we always observed partial rescue of Tr1 cell differentiation

(Fig. 7), and so we sought to find additional regulators that might function downstream of ITK in Tr1 cell development. Blimp-1 has been shown to function jointly with IRF4 in regulating IL-10 production in Foxp3[+] regulatory T cells[27], and is critical for IL-10 production in CD4[+] T helper effector cells[28] as well as antiviral cytotoxic CD8[+] T lymphocytes[29]. In the absence of ITK, CD4[+] T cells cultured under Tr1-polarizing condition exhibited significantly impaired induction of Blimp-1 expression, compared with WT cells (Fig. 8a,b). To determine whether the lack of Blimp-1 expression contributes to the impaired Tr1 cell differentiation in $Itk^{-/-}$ cells, we re-expressed Blimp-1 using retroviral vectors that co-express the human CD2 marker[30] (to identify Blimp-1-expressing cells), and found that over expression of Blimp-1 (Fig. 8c,d) downstream of ITK is unable to rescue Tr1 cell differentiation (Fig. 8e).

**ITK signals via HRas to induce IRF4 during Tr1 cell development.** We and others have found that ITK is required for

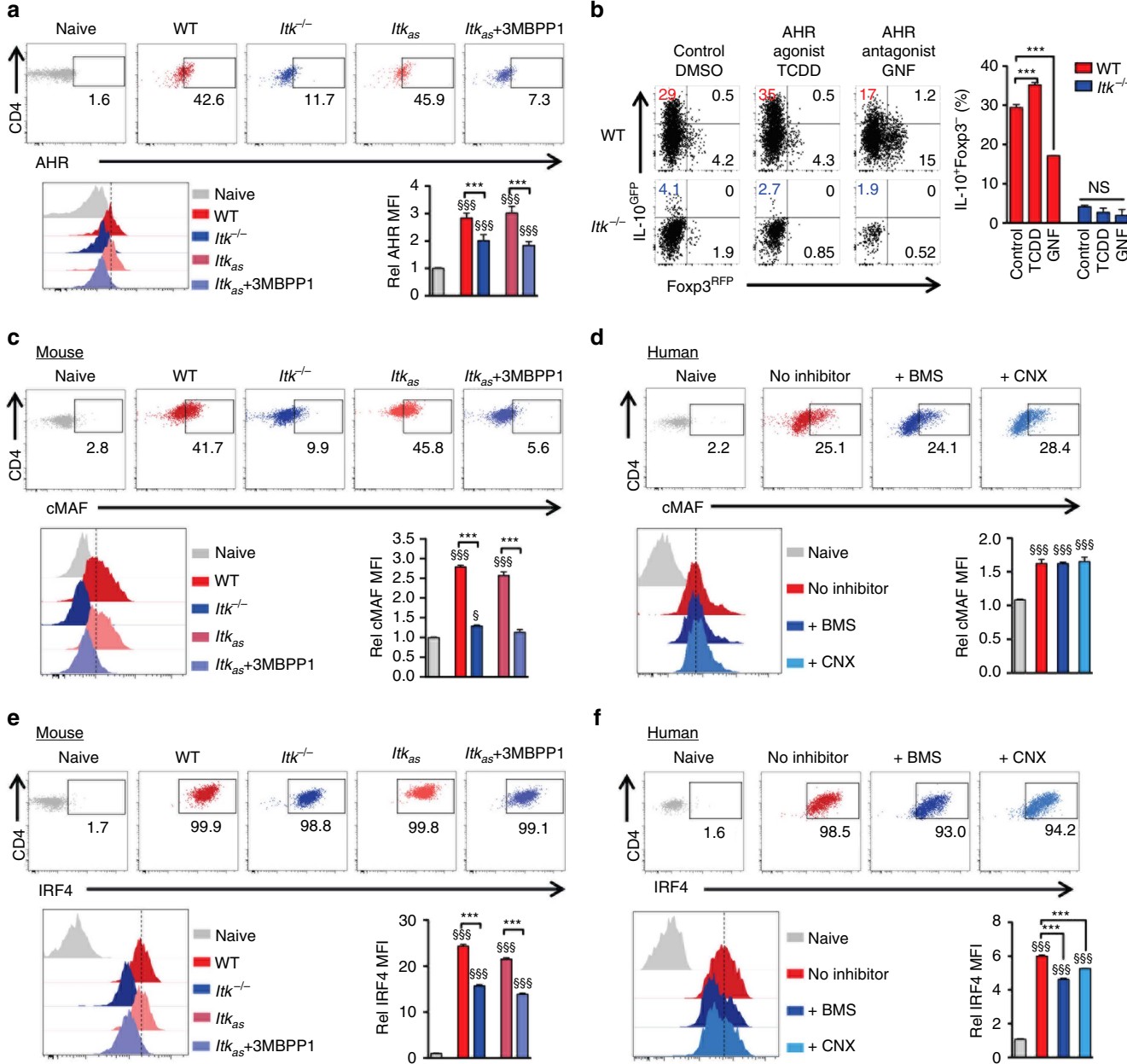

**Figure 6 | ITK regulates IRF4 expression during mouse and human Tr1 cell development.** (**a**) Representative FACS plots and summary of Rel MFI (naive cells level set as 1) of AHR expression in mouse cells cultured under Tr1 polarizing conditions. (**b**) Naive WT and $Itk^{-/-}$ IL-10$^{GFP}$/Foxp3$^{RFP}$ reporter CD4$^+$ T cells were cultured under Tr1 polarizing conditions; AHR agonist TCDD was added to activate AHR, and antagonist GNF was added to inhibit AHR activity. Representative FACS plots of IL-10$^{GFP}$ and Foxp3$^{RFP}$ expression, with summary of percentage of IL-10$^+$Foxp3$^-$ cells are shown. (**c,d**) Representative FACS plots and summary of Rel MFI (naive cells level set as 1) of cMAF expression in (**c**) mouse and (**d**) human CD4$^+$ T cells cultured under Tr1 polarizing conditions. (**e,f**) Representative FACS plots and summary of Rel MFI (naive cells level set as 1) of IRF4 expression in (**e**) mouse and (**f**) human CD4$^+$ T cells cultured under Tr1 polarizing conditions. $n = 6$; data represent results of three experiments in mouse. $n = 4$; data represent results of two experiments in human. Naive CD4$^+$ T cells without stimulation were used as control. $^§P \le 0.05$, $^{§§§}P \le 0.001$, compared with the control level; $^{***}P \le 0.001$; NS, no significance, comparing groups connected, by Non-parametric Mann–Whitney test. Data presented as mean ± s.e.m.

activation of the MAP kinase (MAPK) ERK on TCR engagement in an ITK kinase dependent manner[20,31]. Similarly, MAP kinase JNK activation is also severely impaired in $Itk^{-/-}$ T cells stimulated through the TCR[32]. While TCR signals activate Ras signalling, and the role of Ras in immediate TCR signalling is subtle, Ras is required for Th1 protective immunity during *Leishmania major* infection and Th2-mediated allergic asthma[33] in mouse models. Whether Ras/MAPK signalling plays a role in Tr1 cell differentiation, in particular, downstream of ITK is

unclear. We found that MAPK inhibitors, including ERK, JNK and p38 inhibitors, as well as Ras inhibitor diminished IL-10 induction during Tr1 cell differentiation (Fig. 9). While these inhibitors do not grossly impair T cell activation, they significantly reduced the expression of IRF4 during Tr1 differentiation (Fig. 9).

Retroviral transduction-mediated expression of WT HRas can rescue Tr1 cell development and level of IL-10 expression in the absence of ITK, which is further enhanced by the expression of

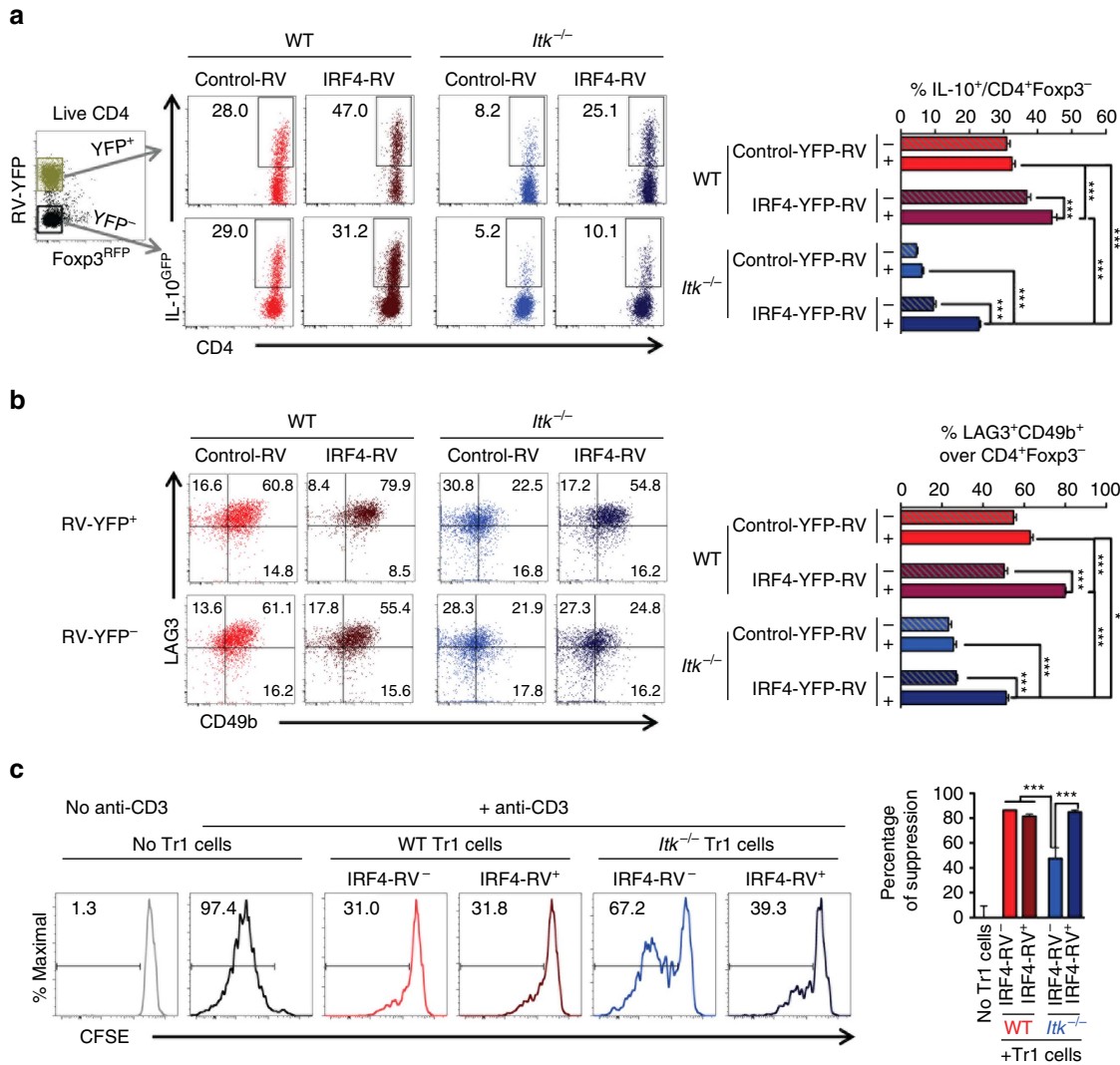

**Figure 7 | Re-expression of IRF4 rescues Tr1 cell differentiation and function in *Itk*$^{-/-}$ cells.** WT and *Itk*$^{-/-}$ naive IL-10$^{GFP}$/Foxp3$^{RFP}$ reporter CD4$^+$ cells were cultured under Tr1 differentiation conditions, and retrovirally transduced with control-YFP or IRF4-YFP. YFP$^+$ (IRF4$^+$) or YFP$^-$ (IRF4$^-$) Foxp3$^{RFP-}$ CD4$^+$ T cells were gated for analysis: representative FACS plots of (**a**) IL-10$^{GFP}$ expression in Foxp3$^-$CD4$^+$ T cells and summary of percentage of IL-10$^+$ cells over live Foxp3$^-$CD4$^+$ T cells; (**b**) LAG3 and CD49b expression and summary of LAG3$^+$CD49b$^+$ Tr1 cells over live Foxp3$^-$CD4$^+$ T cells. $n = 6$. Data represent results of three experiments. (**c**) WT and *Itk*$^{-/-}$ Tr1 cells (CD45.2$^+$, carrying IL-10$^{GFP}$/Foxp3$^{RFP}$ reporters) with or without IRF4-RV were flow sorted and co-cultured with CFSE stained naive CD4$^+$ T cell responders (CD45.1$^+$). Representative FACS plots of CFSE dilution of the responders with or without stimulation in the presence or absence of WT or *Itk*$^{-/-}$ Tr1 cells that are IRF4-RV$^-$ or IRF4-RV$^+$ (Tr1: responder cell ratio = 1:2); and summary of percentage of suppression of the responder proliferation by the indicated Tr1 cells. $n = 6$. *$P \leq 0.05$, ***$P \leq 0.001$, by one-way ANOVA with Tukey's *post-hoc* test. Data presented as mean ± s.e.m.

the constitutively active HRas$^{G12V}$ mutant (Fig. 10a,b). Note that the expression of Ras also rescues the level of IRF4 expression in the *Itk* deficient cells, suggestive of a functional rescue (Fig. 10c).

## Discussion

Our data presented in this report support a requisite role for ITK in Tr1 cell development in both mouse and human. Most importantly, the activity of ITK is required for the suppressive function of Tr1 cells. This function is executed through the kinase activity of ITK, and its down-stream signals through HRas activation and IRF4 expression.

Although both Tr1 cells and Foxp3$^+$ regulatory T cells can produce IL-10 and can suppress T cell proliferation, the role of ITK is different. We and others have reported that ITK, and its kinase activity, suppresses the development of Foxp3$^+$ regulatory

T cells[16,17]. However, our work here indicates that the requirement for ITK in Tr1 cells is more analogous to its role in T effector cells such as Th2, Th9 and Th17 cells[12,15,34]. While the signals downstream of ITK in inducing Th1 differentiation remains unclear, ITK regulation of the transcription factor NFAT has been shown to be critical for its ability to regulate naive cell differentiation to Th2 and Th17 cell fates[12,34]. Furthermore, Schwartzberg and colleagues recently reported that ITK signals via IRF4 to regulate naive cell differentiation to Th9 cell fate and production of IL-9 (ref. 15). Notably, in the absence of ITK, Th9 cell differentiation is fully rescued by the presence of IL-2, production of which is also defective in the absence of ITK. These previous findings suggest that IL-2 may play a role in the function of ITK in Tr1 cell differentiation. Indeed, addition of exogenous IL-2 is able to partially rescue Tr1 cell differentiation in the absence of ITK, although addition of exogenous TGF-β is unable to do so (Supplementary Fig. 4).

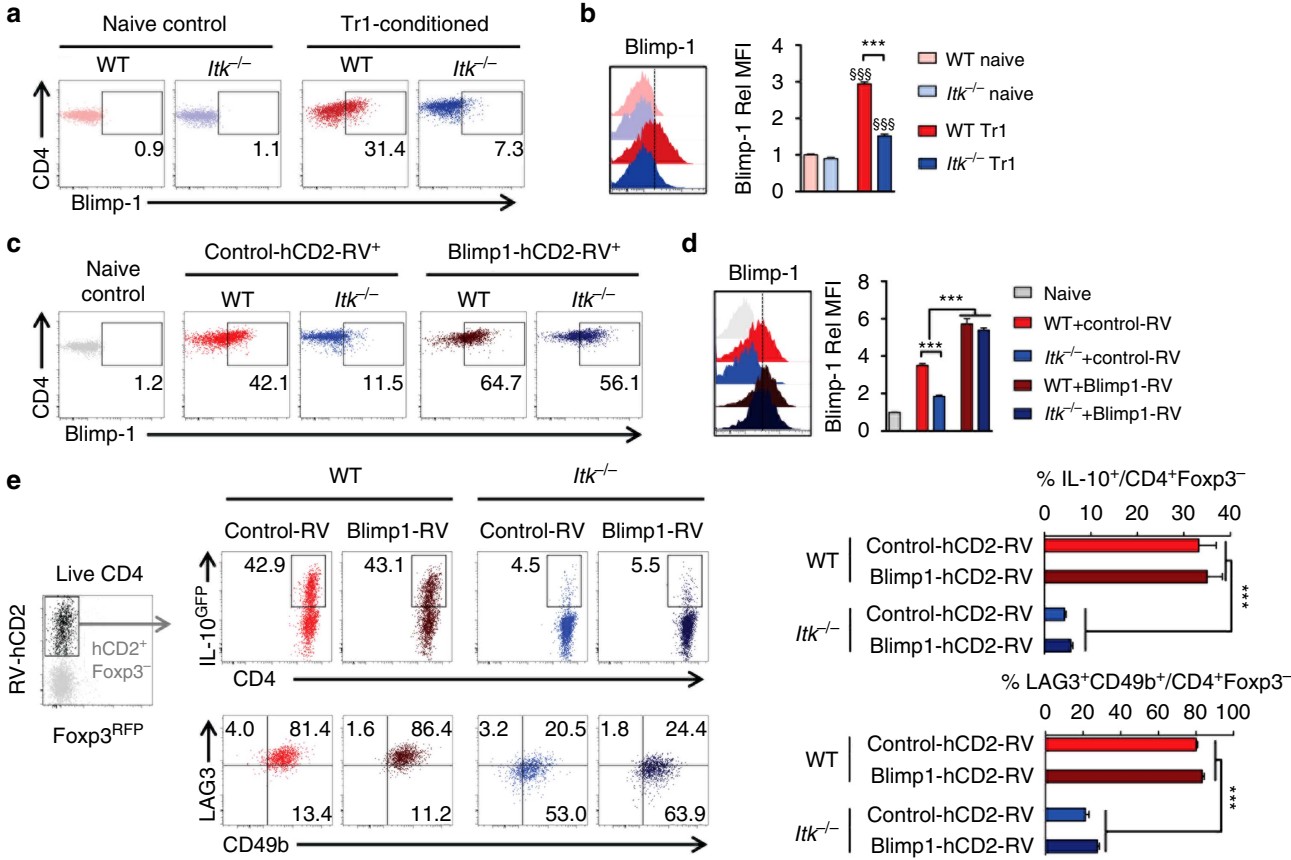

**Figure 8 | ITK regulates but does not depend on the expression of Blimp-1 during Tr1 cell differentiation.** (**a**,**b**) Representative FACS plots and summary of Rel MFI (WT naive cell level set as 1) of Blimp-1 expression in WT and $Itk^{-/-}$ naive CD4$^+$ T cells or CD4$^+$ T cells cultured under Tr1 cell-polarizing condition. $n = 6$. Data represent results of two experiments. $^{\S\S\S}P \le 0.001$, compared with the level in naive cells; $^{***}P \le 0.001$, comparing groups connected, by Non-parametric Mann–Whitney test. (**c–e**) WT and $Itk^{-/-}$ naive IL-10$^{GFP}$/Foxp3$^{RFP}$ reporter CD4$^+$ cells were cultured under Tr1 polarizing condition, retrovirally transduced with control-hCD2 or Blimp-1-hCD2, and analyzed. (**c**) Representative FACS dot plots for the expression of Blimp-1 in RV transduced cells. (**d**) Histogram showing Blimp-1 signal intensity (left) and summary (right) of Rel Blimp-1 MFI. Naive CD4$^+$ T cell population in grey was used as control. $n = 6$. Data represent results of two experiments. $^{***}P \le 0.001$, comparing groups connected, by Non-parametric Mann–Whitney test. (**e**) Representative FACS plots of (top) IL-10$^{GFP}$ expression in Foxp3$^-$CD4$^+$ T cells and summary of percentage of IL-10$^+$ cells over live Foxp3$^-$CD4$^+$ T cells; (bottom) LAG3 and CD49b expression and summary of LAG3$^+$CD49b$^+$ Tr1 cells over live Foxp3$^-$CD4$^+$ T cells. $n = 4$. Data represent results of three experiments. $^{***}P \le 0.001$, by one-way ANOVA with Tukey's post-hoc test. Data presented as mean $\pm$ s.e.m.

We found that the absence of ITK or its kinase activity resulted in defects in expression of the transcription factor AHR, cMAF and IRF4 in T cells activated under Tr1 cell-differentiation conditions. These three transcription factors have all been shown to be important in Tr1 cell differentiation and IL-10 production in T cells [24,25,27,35]. However, an AHR ligand is unable to rescue the Tr1 cell differentiation defects in the absence of ITK, and cMAF expression is not affected in human T cells when ITK kinase activity is inhibited. Our findings that re-expression of IRF4 rescues both Tr1 cell differentiation and function, suggests that IRF4 plays an important role in this process downstream of ITK. AHR can bind to cMAF promoter and regulates cMAF expression during Tr1 cell differentiation[24]. We have also found that re-expression of IRF4 rescued AHR expression in $Itk^{-/-}$ CD4$^+$ T cell under Tr1 cell-differentiating condition (Supplementary Fig. 3c,d). Our data suggest that IRF4 is sufficient to compensate for a lack of ITK as re-expression of IRF4 rescues Tr1 cell differentiation and function. Notably, IRF4 has been reported to lie upstream of Blimp-1, and together both are required for IL-10 production in Foxp3$^+$ regulatory T cells[27]. However, despite the requisite role of ITK signals for the expression of Blimp-1 in Tr1 cells, re-expression of Blimp-1 in ITK deficient cells does not rescue Tr1 cell differentiation. This

suggests that IRF4-regulated genes are critical for this process, and the process cannot be rescued by Blimp-1 alone in the absence of ITK.

We found that inhibition of Ras, and the downstream MAP kinase pathways, particularly ERK and JNK, reduced both Tr1 differentiation and the expression of IRF4. Furthermore, over-expression of HRas, or constitutively active HRas, in $Itk^{-/-}$ T cells also rescued Tr1 cell differentiation, along with IRF4 expression. Given that the re-expression of IRF4 in the absence of ITK rescues Tr1 function, we infer that the HRas pathway, by rescuing IRF4, is able to rescue Tr1 cell function as well. The TCR pathways leading to IRF4 are unclear, and our data shows for the first time that downstream of the TCR and ITK, the Ras/MAPK pathways leads to IRF4 expression during Tr1 cell differentiation.

In addition to conventional differentiation of Tr1 cells from naive CD4$^+$ progenitors, it has recently been shown that Tr1 cells can be derived from cells that had once been Th17 cells but then lost IL-17 production (exTh17 cells) during the resolution of inflammation[11]. Our unique $Itk_{as}$ mice allowed us to be able to generate Th17 cells, where ITK could be specifically targeted by 3MBPP1. We found that similar to the role for ITK in differentiation of Tr1 cells, the kinase activity of ITK is also required for optimal trans-differentiation of exTh17 cell into Tr1

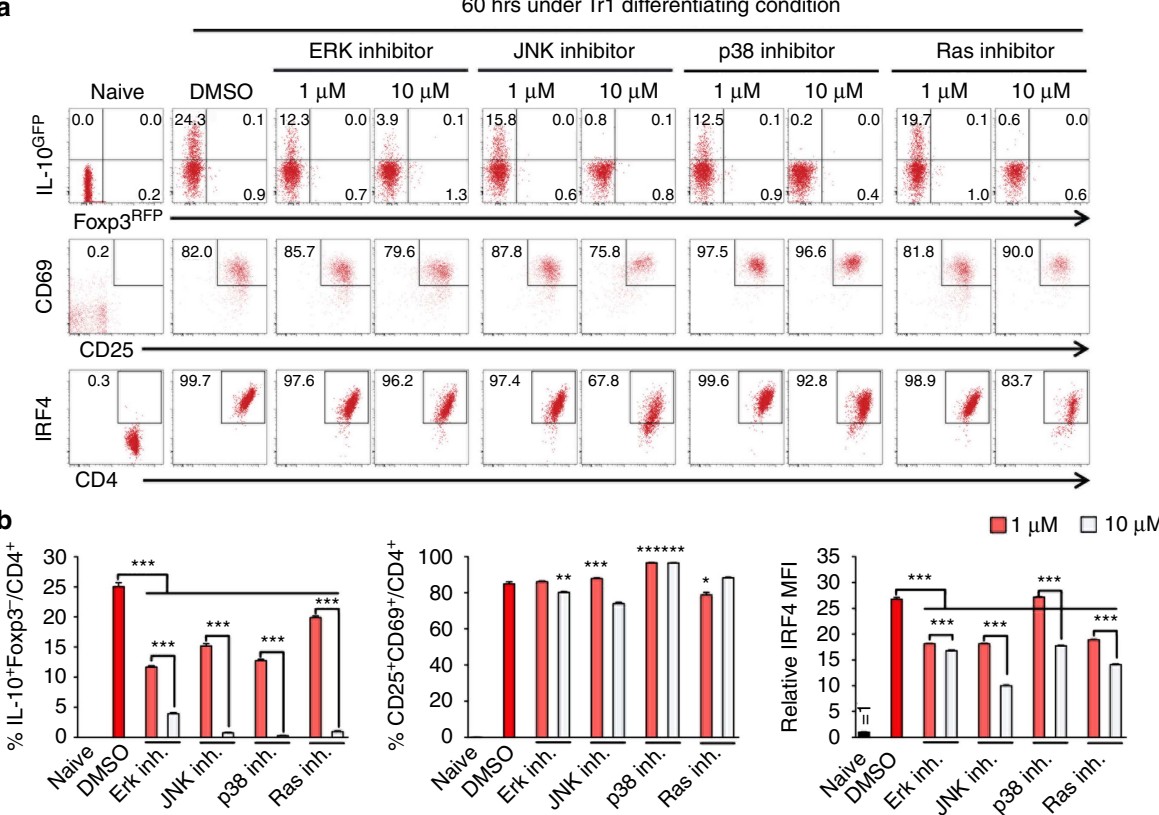

**Figure 9 | Ras/MAPK activity is required for Tr1 cell development and IRF4 expression.** (**a,b**) WT naive IL-10$^{GFP}$/Foxp3$^{RFP}$ reporter CD4$^+$ cells were cultured under Tr1 differentiation conditions, in the presence or absence of ERK1/2 inhibitor (PD098059), JNK inhibitor (SP600125), p38 inhibitor (SB203580), or Ras inhibitor (Kobe0065). Viable CD4$^+$ T cells were analyzed. (**a**) Representative FACS plots of IL-10$^{GFP}$/Foxp3$^{RFP}$, CD69/CD25 and IRF4 expression, and (**b**) summary of percentage of IL-10$^+$Foxp3$^-$ and CD25$^+$CD69$^+$ cells over viable CD4$^+$ T cells, and relative IRF4 MFI (naive cell level set as 1). $n = 6$. Data represent results of two experiments. *$P \leq 0.05$, **$P \leq 0.01$, ***$P \leq 0.001$, by one-way ANOVA with Tukey's *post-hoc* test. Data presented as mean ± s.e.m.

cells (Supplementary Fig. 5b). However, given that ITK kinase activity is also required for Th17 cell development, survival, proliferation and maintenance of RORγt expression under Tr1-polarizing condition (Supplementary Fig. 5a,c), the effect of inhibiting ITK on down-regulation of IRF4 expression and inhibition of IL-10 production might be the result of a combination of cell survival, proliferation and cell fate reprogramming. Whether ITK functions in trans-differentiation of exTh17 cell into Tr1 cells *in vivo* remains to be further explored.

Comparing our results in the mouse and human T cells, we found that in human cells, two different ITK inhibitors exhibit differential effects inhibiting the level of IL-10, cell surface markers LAG3/CD49b and transcription factor IRF4 expression. This may be because of the different characteristics and IC$_{50}$s of these compounds, as well as potentially different off target effects. Note that our work with the 3MBPP1 on the murine cells expressing the *Itk$_{as}$* allele is much more indicative of what would happen when we only inhibited ITK activity, however, we are not able to use this inhibitor in the human T cells for obvious reasons. Furthermore, it is recently reported that IL-10-producing T cells derived from human CD4$^+$ memory T cells exhibit low levels of LAG3/CD49b expression[36], in contrast to what was observed in human Tr1 cells derived from naive human CD4$^+$ T cells[1]. Thus, the effect of inhibiting ITK in human IL-10-producing CD4$^+$ T cells needs to be further explored to consider those derived from naive precursors versus memory cells.

In our experiments, we also examined the responses of *Itk$^{-/-}$* mice to infection with *N. brasiliensis* and Flu. While the response of *Itk$^{-/-}$* mice to infection with *N. brasiliensis* has been previously reported[34], this work was performed before the discovery of Tr1 cells, and the response of these mice to infection with Flu have not been reported. In both infection models, we found that the Tr1 cell response was quite organ specific, with significant Tr1 cell response found largely in the lungs and to a lesser extent in the mesenteric/draining lymph nodes in both models, with little response in the spleen (Supplementary Fig. 6). Furthermore, while infection with *N. brasiliensis* led to increased recovery of parasites in the absence of ITK (Supplementary Fig. 6), there was no difference in the recovery of virus from the lungs of Flu infected mice, and no difference in body weights, although there was reduced survival of *Itk$^{-/-}$* mice (Supplementary Fig. 6). The complex immunological phenotype of the *Itk* deficiency beyond our analysis of Tr1 cell responses *in vivo*, with multiple cell types exhibiting different phenotypes[12–17,23,26,32,34,37–46], makes it more challenging to provide an explanation for these phenotypes without significant additional experiments.

Regulatory T cell function can be a double-edge sword, beneficial in controlling inflammation and promoting immune tolerance, but detrimental in causing immune depression under conditions of infections or in the face or tumours. Tr1 cells have been found to be abundant in chronic hepatitis C virus[47] and *Mycobacterium tuberculosis*[48] infection, as well as in tumour microenvironments[49,50], which may lead to a lack of or reduced

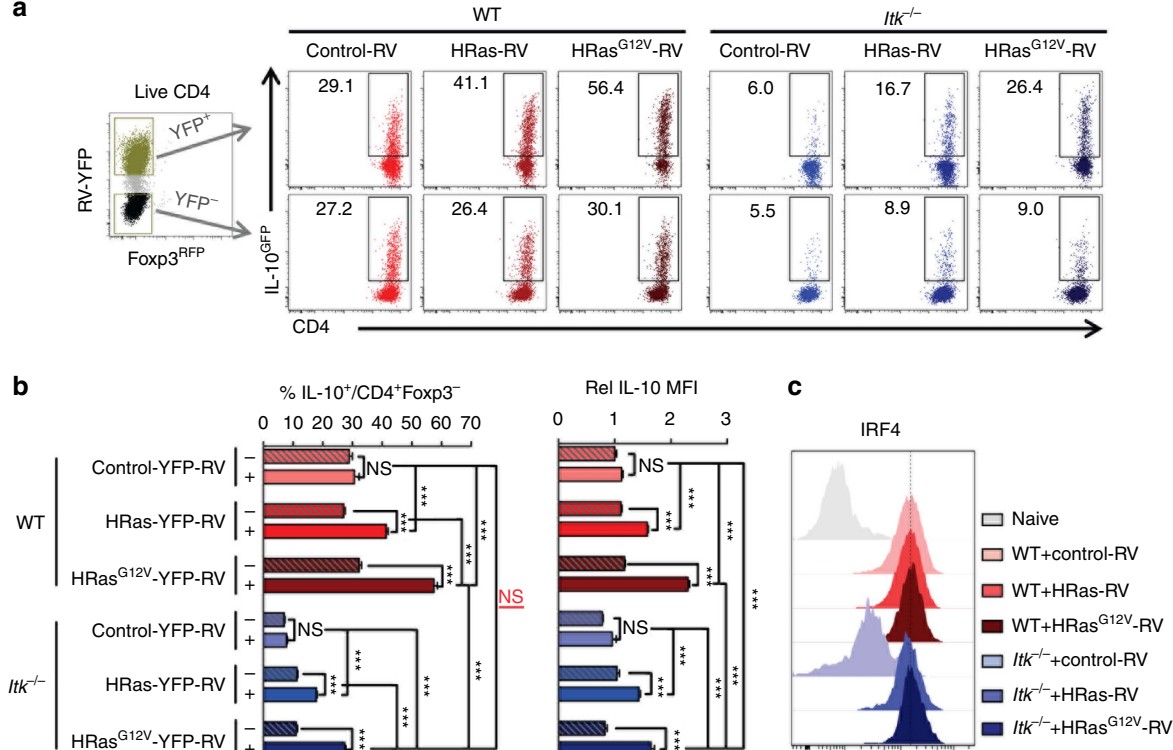

**Figure 10 | HRas activity rescues Tr1 cell development and IRF4 expression in *Itk*$^{-/-}$ cells.** WT and *Itk*$^{-/-}$ naive IL-10$^{GFP}$/Foxp3$^{RFP}$ reporter CD4$^+$ cells were cultured under Tr1 differentiation conditions, retrovirally transduced with control-YFP, WT HRas-YFP or its constitutively active mutant HRas$^{G12V}$-YFP. (**a**) Representative FACS plots of IL-10$^{GFP+}$ T cells. (**b**) Summary of percentage of IL-10$^{GFP}$ cells over viable CD4$^+$Foxp3$^-$ T cells and relative IL-10 MFI in IL-10$^+$Foxp3$^-$ cells (WT control level set as 1). (**c**) Representative FACS plots of IRF4 expression in RV-transduced cells. Naive CD4$^+$ T cells were used as control in grey. n = 6. Data represent results of three experiments. *$P \leq 0.05$, ***$P \leq 0.001$; NS, no significance, by one-way ANOVA with Tukey's *post-hoc* test. Data presented as mean ± s.e.m.

immune response to clearing viruses, bacteria or tumours. Given the functional requirement for ITK kinase activity in differentiated Tr1 cells, targeting ITK activity and its signalling pathway specifically in Tr1 cells may be a promising strategy in modulating immune suppression due to the functions of Tr1 cells under such circumstances, although further work in humans will have to be done to determine this. These findings place ITK and its signalling axis as promising therapeutic targets to modulate Tr1 cell numbers and activity for treating diseases such as inflammation, chronic infection and tumours.

## Methods

**Mice.** All mice were on the C57BL/6 background. CD45.1 (B6.SJL-*Ptprc*$^a$ *Pepc*$^b$/BoyJ; 002014), *Rag1*$^{-/-}$ (B6.129S7-*Rag1*$^{tm1Mom}$/J; 002216), IL-10$^{GFP}$ (B6(Cg)-*Il10*$^{tm1.1Karp}$/J; 014530)[51] and Foxp3$^{RFP}$ (C57BL/6-*Foxp3*$^{tm1Flv}$/J; 008374)[52] reporter mice were purchased from the Jackson Laboratory (Bar Harbor, ME). IL-17A$^{GFP}$ (B-IL17-EGFP KI; available at Jackson Laboratory as C57BL/6-*Il17a*$^{tm1Bcgen}$/J; 018472) reporter mice were from Biocytogen (Worcester, MA)[53]. Reporter strains were crossed to generate IL-10$^{GFP}$/Foxp3$^{RFP}$ or IL-17A$^{GFP}$/Foxp3$^{RFP}$ dual reporter strains in WT, *Itk*$^{-/-}$ or *Itkas* background. ITK*as/Itk*$^{-/-}$ (Tg(hCD2-*Itkas*)*Itk*$^{-/-}$) mice, that is, *Itkas*, were as previously described[16,20], and harbour an altered ATP binding pocket (F434G/Δ429) in ITK kinase domain allowing specific inhibition by 1-(*tert*-Butyl)-3-(3-methylbenzyl)-1H-pyrazolo[3,4-d]pyrimidin-4-amine (3MBPP1). All mice were bred in house, and both female and male mice were used between 5 and 12 weeks of age. All experiments were approved by the Office of Research Protection's Institutional Animal Care and Use Committee at Cornell University.

**Antibodies and other reagents.** All fluorescent antibodies are listed in 'fluor-ochrome-target (clone; annotation if desirable)' format. Mouse antibodies include purified anti-CD16/32 (93; Fc block), CD3ε (145-2C11), CD28 (37.51), IFN-γ (XMG1.2), and IL-12 (C17.8) antibodies were from BioLegend (San Diego, CA); Brilliant Violet 421-ICOS (C398.4A; human/mouse), FITC-TCRβ (H57-597),

PE-IL-10 (JES5-16E3), APC-LAG3 (C9B7W), Alexa Fluor 647-Blimp-1 (5E7), PE-Cy7-CD49b (HMα2), PE-Cy7-CD62L (MEL-14) and Brilliant Violet 785-PD-1 (29F.1A12) from BioLegend; eFluor 450-CD4 (GK1.5), PE-AHR (4MEJJ), PE-cMAF (sym0F1; human/mouse), eFluor 660-IRF4 (3E4; human/mouse), Alexa Fluor 700-CD45.2 (104), Alexa Fluor 700-CD4 (GK1.5), PerCP-Cy5.5-CD25 (PC61.5), PE-Cy7-CD45.1 (A20) and PE-Cy7-Ki67 (SolA15; human/mouse) from eBioscience; BD Horizon BV421-RORγt (Q31-378), BD Horizon V500-CD44 (IM7), Alexa Fluor 488-IL-4 (11B11), PE-CD44 (IM7), PE-CD69 (H1.2F3) and APC-Cy7- TCRβ (H57-597) from BD Biosciences. Human antibodies include purified anti-CD3ε (OTK3) and CD28 (28.2) were from eBioscience; PE-IL-10 (JES3-19F1), Alexa Fluor 647-LAG3 (11C3C65), Brilliant Violet 785-PD-1 (EH12.2H7); FITC-CD4 (OKT4) and APC-FOXP3 (236A/E7) from eBioscience; FITC-CD49b (AK-7) and Alexa Fluor 700-CD4 (RPA-T4) were BD Biosciences. Human TruStain FcX (Fc receptor blocking solution) was from Biolegend; cell proliferation dye eFluor 450 and fixable viability dye eFluor 506 were from eBiosciences. Antibodies used in cell cultures or *in vivo* were in the concentration as indicated, and those for flow staining were used at 1–2 µg ml$^{-1}$ or as instructed by the manufacturer when concentration was not available.

**Cell isolation from organs.** Blood cells were collected through cardiac puncture, and red blood cells were lysed before analysis; lungs were minced and digested in 0.2 mg ml$^{-1}$ Liberase TL (Sigma, St Luis, MO) in 37 °C for 30 min, then filtered and red blood cells were lysed before analysis; intestines were flushed, opened long-itudinally and inner contents were removed with the blunt end of the scissors, then cut into 0.5-cm fragment, followed by digestion in 100 U ml$^{-1}$ collagenase VIII (Sigma) in 37 °C for 1 h, filtered, and lymphocytes were isolated using gradient separation by 40 and 80% Percoll (GE Healthcare, Wilkes-Barre, PA) solutions; perigonadal adipose tissues were minced and digested in 500 U ml$^{-1}$ collagenase I (Worthington Biochemical Corp., Lakewood, NJ) in 37 °C for 30 min, filtered and red blood cells were lysed before analysis. A total of 50–150 U ml$^{-1}$ DNase I (Sigma) were added during digestion to reduce cell death triggered by free DNA.

**Mouse Tr1 differentiation.** To induce Tr1 cells *in vivo*, WT and *Itk*$^{-/-}$ Foxp3$^{RFP}$IL-10$^{GFP}$ dual reporter mice were injected with 15 µg per mouse anti-CD3ε (145-2C11) intraperitoneally on day 0 and 2, and analyzed on day 4, as previously described[3]. To induce Tr1 cells *in vitro*, CD4$^+$CD8$^-$TCRβ$^+$

Foxp3$^{RFP-}$ CD44$^-$CD122$^-$ thymic or CD44$^-$CD62L$^+$ splenic naive CD4$^+$ T cells were sorted on BD FACS Aria II or Fusion systems (BD Biosciences, San Jose, CA), then cultured with Mitomycin-C (Sigma, 50 μg ml$^{-1}$) treated antigen-presenting cells (APC; $Rag^{-/-}$ splenocytes) at 1:2 ratio in the presence of anti-CD3ε (1 μg ml$^{-1}$), anti-CD28 (1 μg ml$^{-1}$), recombinant murine (rm) IL-27 (Peprotech, 20–25 ng ml$^{-1}$), anti-IFN-γ and anti-IL-12 (5–10 μg ml$^{-1}$). Murine Tr1 cells induced by this protocol produce low levels of IFN-γ (Supplementary Fig. 2a) and IL-4 (Supplementary Fig. 7) as determined by flow cytometry. To test the effect of IL-2 and TGF-β in Tr1 cell differentiation, recombinant human (rh) TGF-β or IL-2 (Peprotech) were added as indicated. ITK inhibitor BMS-509744 (Millipore) was used at 1 μM (ref. 22); ITK$_{as}$ specific inhibitor 3MBPP1 (Millipore) was used at 2 μM (ref. 16). AHR agonist 2,3,7,8-tetrachlorodibenzo-p-dioxin (TCDD) and antagonist GNF-351 (kind gifts from Dr G. Perdew, Penn State University) were used at 200 nM (ref. 54). MAPK inhibitors PD098059 (targeting ERK pathway)[55,56], SP600125 (targeting JNK pathway)[57], SB203580 (targeting p38 pathway)[58] and Ras inhibitor Kobe0065 (ref. 59) were all from Sigma and used at 1 or 10 μM as indicated.

**Human Tr1 differentiation in vitro.** Leukopaks were procured from the New York Blood Center (Long Island, NY) collected from healthy cohorts. The use of human blood cells was considered exempt and approved by the Institutional Review Board at Cornell University. Human peripheral blood monocytes (PBMCs) were isolated from blood using gradient separation in Ficoll-Paque PLUS (GE Healthcare). PBMC were cultured in full RPMI-1640 medium (RPMI-1640 medium base (Gibco, 11875-093) with 10% fetal bovine serum (ATLAS Biological, F-0500-D), 1 mM sodium pyruvate (Gibco, 11360-070), 2 mM L-glutamine (Gibco, 25030-081), 1 mM non-essential amino acid (Gibco, 11140-050), 5 mM HEPES (HyClone, SH30237.01), and 100 U ml$^{-1}$ penicillin/streptomycin (Gibco, 15140-122)) for 30 min in 37 °C, then non-adherent cells were used to enriched CD4$^+$ T cells using anti-human CD4 microbeads (Miltenyl Biotec, San Diego, CA), followed by flow cytometric sorting to purify CD4$^+$TCRβ$^+$ CD45RO$^-$ naive cells (purity >98%), and adherent cells were treated with Mitomycin-C (Sigma, 50 μg ml$^{-1}$) in 37 °C for 30 min and used as APCs. Anti-human CD3ε (1 μg ml$^{-1}$) and CD28 (CD28.2, eBioscience, 1 μg ml$^{-1}$), recombinant human (rh) IL-2 (PeproTech, 50 U ml$^{-1}$), IL-15 (PeproTech, 1 ng ml$^{-1}$), IL-10 (PeproTech, 100 U ml$^{-1}$), and IFN-α2b (R&D System, 5 ng ml$^{-1}$) were added to differentiate human Tr1 cells. Selective ITK inhibitor BMS-509744 was used at 1 μM (ref. 22); broad Tec family kinase inhibitor CNX584 (targeting both ITK and BTK) was used at 50 nM (ref. 23).

**Mouse Th2 and Th17 differentiation in vitro.** CD4$^+$TCRβ$^+$Foxp3$^{RFP-}$ CD44$^-$CD62L$^+$ splenic naive CD4$^+$ T cells were isolated from IL-17$^{GFP}$/Foxp3$^{RFP}$ or IL-10$^{GFP}$/Foxp3$^{RFP}$ mice through sorting on BD FACS Aria Fusion system, then cultured for 3 days with Mitomycin-C treated APCs at 1:3 (T:APC) ratio in the presence of various antibodies and/or cytokines as indicated below: Th2-polarizing condition included anti-CD3ε (1 μg ml$^{-1}$), anti-CD28 (1 μg ml$^{-1}$), rh IL-2 (PeproTech, 20 ng ml$^{-1}$), and rm IL-4 (PeproTech, 40 ng ml$^{-1}$), anti-IFN-γ and anti-IL-12 (10 μg ml$^{-1}$), as we previously described[14]; and Th17-polarizing condition included anti-CD3ε (1 μg ml$^{-1}$), anti-CD28 (3 μg ml$^{-1}$), rm IL-6 (R&D systems, 50 ng ml$^{-1}$), rh TGF-β (PeproTech, 1 ng ml$^{-1}$), anti-IL-4, anti-IFN-γ and anti-IL-12 (5–10 μg ml$^{-1}$), as we previously described[12].

**N. brasiliensis and Influenza A/WSN/1933 (WSN) infection.** Mice were given 500 L3 N. brasiliensis larvae per mouse through subcutaneous injection, or 1 LD$_{50}$ (10$^4$ PFU) WSN per mouse through intranasal treatment. Cells from the spleen, lymph nodes and lungs were analyzed at the indicated time points. Parasite burden in mice infected with N. brasiliensis was determined on 7 days post infection (7 d.p.i.) by opening the small intestines longitudinally under an optical microscope. Weight of mice infected with WSN was collected every 24 h, and mice that lost >20% of the original weight were killed and recorded as dead. Plaque-forming units (PFU) of WSN in lungs of mice infected with WSN were determined by plaque assay using MDBK cells[60]. Lungs were sectioned and stained with Periodic Acid-Schiff (PAS) by the Animal Health Diagnostic Center at Cornell University.

**In vitro suppression assay.** CD45.1$^+$CD45.2$^-$ naive responder cells were labeled with 5 μM CFSE (Thermo Fisher Scientific), and co-cultured with Mitomycin-C treated APCs, in the presence or absence of anti-CD3ε (0.5–1 μg ml$^{-1}$). WT and Itkas Tr1 cells (CD45.2$^+$) were differentiated in vitro, sort purified (> 98% IL-10$^{GFP+}$Foxp3$^{RFP-}$), and used as suppressors in the indicated ratios to the naive responders (CD45.1$^+$). 3MBPP1 (1–2 μM) was added to specifically inhibit the kinase activity of ITK in Itk$_{as}$ Tr1 cells, without affecting the kinase activity of ITK in responder T cells. Division of responder cells was analysed 3 days later, and Division Index (DI) was calculated using the FlowJo (Tree Star) 'Proliferation Platform', in which DI is defined as the ratio of 'the total number of divisions/the number of cells at start of culture'. Then the average of DI of responder cells with anti-CD3 stimulation but no Tr1 cell suppression was termed DI$_0$, while the DI of responder cells under condition $i$ was termed as DI$_i$. The percentage of suppression (PS) under condition $i$ was defined as 'PS$_i$ = 100 • (1−DI$_i$/DI$_0$)'[61].

**Retroviral transduction.** The pGC-IRES-yellow fluorescent protein retroviral vector (YFP-RV), pGC-IRF4-IRES-YFP-RV, and packaging vector pCL-Eco are as previously described[62] (kind gifts from Dr A. Pernis, Hospital for Special Surgery). To generate pGC-HRas-IRES-YFP-RV and pGC-HRas$^{G12V}$-IRES-YFP-RV, the HRas and HRas$^{G12V}$ cDNA (kind gift from Dr R. Cerione, Cornell University) were cloned into the pGC-IRES-YFP-RV via the 5′ Not I and 3′ Xho I restriction site ligation respectively, and verified by DNA sequencing. The pMX-IRES-human CD2 (hCD2) RV and pMX-Blimp1-IRES-hCD2 RV are as previously described[30] (kind gifts from Dr T. Malek, University of Miami). HEK 293FT cells were cotransfected with the retroviral vector and the packaging vector. Retroviral supernatants were harvested 48 and 72 h post transfection, concentrated using Retro-X concentrator (Clontech, Takara Bio, Mountain View, CA), and used to infect naive CD4$^+$ T cells that had been stimulated under Tr1 cell differentiating conditions overnight. 48 h after infection, CD4$^+$ T cells that expressed the protein encoded by the retroviral vector (YFP$^+$ or hCD2$^+$) were analysed for Tr1 cell differentiation and function.

**Flow cytometry.** Surface staining of live cells were done in the presence of Fc block and fixable viability dye. For intracellular cytokine staining, cells were stimulated with PMA (100 ng ml$^{-1}$), Ionomycin (0.5 μM), Brefeldin A (5 μg ml$^{-1}$) and Monensin (2 μM) (Sigma-Aldrich, St. Louis, MO) for 4 h, followed by fixation with 2% paraformaldehyde (Electron Microscopy Sciences, Hatfield, PA), permeabilization and staining with anti-cytokine antibodies in PBS/0.15% saponin (Sigma). Staining for nuclear factors Ki67, FOXP3, AHR, cMAF, Blimp-1 and IRF4 were performed with Foxp3 staining buffer kit (eBioscience). Flow cytometry data were acquired on LSRII, FACS Aria II or Fusion systems (BD Biosciences), and analyzed in FlowJo (Tree Star, Ashland, OR). All analyses were done gated on fixable viability dye negative singlet population. Representative plots of gating strategies for sorting and cell analyses are shown in Supplementary Fig. 8.

**Quantitative real time PCR.** Total RNA was extracted using the RNeasy kit (Qiagen) and cDNA was synthesized using the iScript cDNA synthesis kit (Bio-Rad). To quantify the gene transcript levels, 'Best Coverage' gene probes for mouse Gapdh (Mm99999915_g1), mouse Il10 (Mm01288386_m1), human GAPDH (Hs02786624_g1) and human IL10 (Hs00961622_m1) (TaqMan, Life Technologies, Grand Island, NY) were used on a ViiA 7 real-time PCR system following the manufacturer's recommended program (Hold Stage: 50 °C for 2 min, 95 °C for 10 min; then PCR Stage with 40 cycles: 95 °C for 15 s and 60 °C for 1 min). IL-10 relative mRNA level was calculated by normalizing Il10 values to the internal loading control Gapdh values first, and then to the average of Il10/Gapdh values in WT.

**Statistical analysis.** Non-parametric Mann–Whitney test, one-way ANOVA with Tukey's Post-Hoc test, two-way ANOVA, and Log-rank test were performed using GraphPad Prism v5.00 (GraphPad, San Diego, CA), with $P \leq 0.05$ considered statistically significant. 'NS' refers to 'No Significance'.

**Data availability.** The data that support the findings of this study are available from the corresponding author on request.

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

## Acknowledgements

We thank A. Redko for animal care and L. Zhang for technical assistance; D. LaTocha, C. Bayles, Dr R. Williams and R. MacDonald for help in flow sorting; Dr E. Tait Wojno for N. brasiliensis and helpful discussion; Dr D. Topham for influenza A virus; Dr G. Perdew for AHR agonist and antagonist; Drs A. Pernis and S. Gupta (pCL-Eco and pGC) and T. Malek and A. Yu (pMX) for retroviral and packaging plasmids; Drs R. Cerione and M. Antonyak for HRas cDNA templates; Drs M. Straus, V. Tse and G. Whittaker for MDBK cells and help with the WSN plaque assay; and Drs D. Russell and D. Gludish for HEK 293FT cells. This work was supported in part by grants from the National Institutes of Health (AI108958, AI120701 and AI126814 to A.A.), a Careers in Immunology Fellowship from the American Association of Immunologists (to W.H.), and the Talent Program of The Third Affiliation Hospital of Sun Yat-sen University (555 to W.H. and S.-G.Z.).

## Author contributions

W.H. and A.A. conceived research, designed experiments, analysed and interpreted data, and wrote the manuscript; W.H., S.S. and N.K. performed experiments; S.-G.Z. contributed reagents and intellectual input.

## Additional information

**Competing interests:** The authors declare no competing financial interests.

