## [Peer Review File · Nature Communications]

Reviewers' comments:

Reviewer #1 (Remarks to the Author):

Summary: This study identified a novel role for ITK signalling through IRF4 for the differentiation and function of induced Tr1 cells from mice and humans. The main comparisons were performed between induced Tr1 cells from FOXP3/IL-10 dual reporter mice, either WT or ITK KO. The authors used several experimental methods and also studied induction of human Tr1 cells from healthy controls in the presence or absence of ITK inhibitors.

General comments: The area of investigation is novel and the authors have presented sufficient data to support their claims on the role of ITK and IRF4 in Tr1 cell induction. However, the study largely relies on data from induced Tr1 cells, meaning the role for ITK signalling pathways in *in vivo* differentiated Tr1 cells, particularly human cells, remains underexplored. The induced-Tr1 cells cannot be called Tr1 cells without a full phenotyping and cytokine secretion profile being shown.

The levels of IL-10 production are only assessed at the protein level in humans, with one mouse experiment looking at mRNA levels, the authors should also assess changes in the levels of IL-10 mRNA transcripts in human cells, to determine if the inhibition of ITK is reducing IL-10 expression at both the transcriptional and translational level.

As ITK is described to be important in TCR signalling is the phenotype of ITK $-/-$ cells a result of these cells not being as efficiently activated? Do the KO cells upregulate activation markers to the same extent? It is stated that they proliferated equally but data is not shown - this is important and should be included. At the first introduction of ITK it should be named in full and discussed in more detail, a comment in the introduction on known differences in downstream signals of TCR/ITK would help contextualise the data. It should also be briefly described how efficient the ITK KO model is i.e. what is the percent residual kinase activity?

Specific comments:

Only a few gating strategies are shown for isolation of starting cell populations; these should be included as supplemental data for all FACS analysis/sorting approaches. It is also unclear whether viability staining was performed in all experiments and whether dead/dying cells have been excluded from analyses/sorting.

In the methods it is stated that for *in vitro* differentiation of mouse Tr1 cells there was varying amounts of IL-27 (20-25ng/mL), anti-IFN γ , anti-IL-12 (5-10ug/mL) added and TGF β was sometimes added. Also that analysis was performed 48-72h later. These are large variations and without knowing exactly which culture conditions/time of induction was used in each experiment it is hard to compare the data, particularly as cytokine profiles can vary greatly from 48-72h - the method used to generate data in each experiment needs to be clear. Why was this time (48-72h) chosen as a read out? Does the IL-10 profile change over longer period of *in vitro* culture/with repeated TCR stimulation?

The culture conditions for mice Tr1 cells is quite different to that used for human Tr1 cell differentiation where IL-2, IL-15 and IL-10 are added. It has been shown that some Itk $-/-$ effects on Th polarization is due to altering IL-2 sensitivity, does adding these additional cytokines to mouse cultures affect the results?

Fig 1A. The phenotype of the generated dual reporter ITK^{-/-} should be briefly described, do they develop any spontaneous autoimmunity, have defects on any other cell types. Do they have the expected phenotype of increased Th1 and Tregs and decreased Th17, Th2 and Th9? The bar graphs showing %Tr1 - what is this a percent of? How are induced Tr1 cells defined, seems to be only by IL-10 expression, without a more extensive phenotype shown of the IL-10⁺ cells this is insufficient. Given this is such a drastic phenotypic difference it would be good to also show by direct protein staining and at mRNA level for IL-10 (and IL-4 as true Tr1 cells should be IL-4^{neg}).

Fig 1B. The authors focus on the ability to induce Tr1 cells from the ITK KO mice, although it appears that there is no difference in the levels of circulating in vivo derived IL-10⁺ CD4⁺ T cells, this is not discussed (levels in spleen actually look higher in KO mice!). It would be interesting to see if the same phenotype of reduced cMAF, AhR and IRF4 is evident in the IL-10⁺ cells isolated from KO mice vs WT without any manipulation. Title '% of Tr1' appears to be a typo.

Fig 1C. I am glad to see these data showing pathogen-driven in vivo generation of IL-10 producing cells, this is important, although again a full phenotype should be shown to call them Tr1 cells. Why were the lungs isolated for analysis of CD4⁺ T cells following helminth infection? Are the same differences seen in the MLN, peripheral blood, spleen etc as per 1B?

Fig 1D. Interestingly the viral infection is driving more IL-10 induction than the helminth infection in 1C, perhaps the authors could comment on this and also that there doesn't appear to be a significant affect on the ability of FOXP3⁺ cells to produce IL-10 in the ITK^{-/-} mice. Is this defect in IL-10 induction restricted to the lung or generally seen throughout the mice?

Fig 2A. Statistical differences between Itk^{-/-} and Itkas+3MBPP1 should be shown for all graphs.

Fig 2B. The expression levels of the Tr1 markers are only shown separately, what is the percent of LAG3⁺CD49b⁺ cells within IL-10⁺ cells? Dot plots should be shown alongside bar graphs and must include the gating strategy showing how positive gates were set.

Fig 2C. Is the same effect seen with TCR re-stimulation as opposed to PMA/Ionomycin stimulation? Statistical differences between all populations should be shown.

Fig 2D. Same comment as for mice that full Tr1 phenotype need to be shown in human cells.

Fig 2E. Dot plots showing gating are required and an analysis of %LAG3⁺CD49b⁺ cells within IL-10⁺ cells. Suppression assays should be performed with human induced 'Tr1 cells' to confirm a regulatory phenotype.

Fig 2F. A better demonstration of trans-differentiation would be to isolate in vivo differentiated Th17 cells and culture in Tr1 conditions as the induced Th17 probably do not represent a truly committed Th17 cell population. What was the amount of IL-10 (and other cytokines/Tr1 markers) expression in cells in Th17 condition prior to sorting and culture in Tr1 condition? Do they also upregulate Tr1 markers along with IL-10 expression following culture in Tr1 conditions. If the IL17/FOXP3 reporter mice are infected as per Fig 1 do they also have reduced trans-differentiation in vivo? This should also be shown using ex vivo isolated human Th17 cells. Currently there is insufficient data in this figure to conclude that ITK affects trans-differentiation of committed Th17 cells into Tr1 cells.

Fig 3. It is unclear what 'background' histogram represents. Again, dot plots showing gates are required for all data only shown in histogram or bar plots. Are the differences in cMAF due to differences in culture conditions between mice and humans?

Fig 3G. It is stated that IRF4 re-expression rescued Tr1 cell development, this appears to only be a partial rescue and text should be amended to reflect percent rescue and again is entire phenotype

rescued or only IL-10 expression? Show stats between *Itk*^{-/-} and WT populations on bar graph.

This finding for IRF4 seems to be a key piece in the Tr1 puzzle. Do the authors have data on the relative expression of IRF4 in WT 'Tr1 cells' compared to Tregs and other Th cells to show that this is a feature of Tr1 cells? It is disappointing that suppression assays were not done with the IRF4 transduced cells to see if IRF4 re-expression can also rescue suppressive function.

Fig 4. Gating needs to be shown for assay analysis, in particular were dead responders excluded from analysis and was %death different between conditions. Rather than % proliferation % suppression should be shown, calculated using the division index as per McMurchy and Levings 2012 EJI. It is insufficient to only show suppression assays at a 1:1 ratio, a range of suppressor: responder cells to at least a 1:16 ratio should be shown. It would have been nice to see a direct investigation of the importance of IL-10 secretion for suppressive function through either transwell assays, adding IL-10 bAb or IL-10R nAb or by adding back recombinant IL-10 to the *Itk*^{+/+}+3MBPP1 wells to same levels seen in WT Tr1 well to see if suppression could be restored.

It should be made clear in the discussion that the role for *Itk* in Tr1 development has mainly been shown in vitro and it is unexplored whether it is required for in vivo differentiation in humans.

Minor comments:

The PCR cycling conditions and machine used should be provided.

In some figure legends the number of mice/patients used for each experiment and the number of experiments that are pooled are unclear - fix throughout. It is not enough to state $n > x$, the exact n should be stated.

Most statistical analysis used Student's *t* tests were used, was a test of normality done to determine normal distribution of the data sets being analysed? Otherwise the non-parametric Mann-Whitney test is a more suitable test.

Reviewer #2 (Remarks to the Author):

This paper by Huang et al, "ITK signaling via IRF4 regulates the development and function of type 1 regulatory T cells" reports a mechanism by which ITK signaling under TCR stimulation regulates Tr1 development in vitro and in vivo. The authors report that IL-10 production by FoxP3⁺ T cells require ITK signaling in several systems. *Itk* deficient mice or ITK inhibitor showed reduction of the frequency and number of IL-10⁺Foxp3⁺ T cells (Tr1 cells) in various organs following TCR activation, and in mucosal system during parasitic and viral infections. They also showed ITK signaling is important for regulating balance between IFN γ and IL-10 in Tr1 differentiation and trans-differentiation of Th17 into Tr1 cells. Furthermore, they showed a molecular mechanism by which IRF4 rescue Tr1 cell differentiation in *Itk* deficient cells. They also performed a suppression assay by which *Itk* signaling is important for Tr1 function in vitro. Thus, the authors conclude that TCR/ITK/IRF4 pathway is required for functional development of Tr1 cells.

Major critique:

The data presented in the paper are clear, that ITK signaling under TCR stimulation is important for Tr1 cell development and function, however the importance of TCR signaling for T cell differentiation is expected and therefore detract somewhat from novelty of the observations unless they show specificity of this cascade compared to other TCR signaling cascades.

Furthermore, they did not show which protein is phosphorylated by ITK and binds on IRF4 transcription promoting region. Current analysis is not sufficient for proving this molecular mechanism of TCR/ITK/IRF4 pathway.

Specific critique:

1. In Figure1, the authors are claiming ITK signaling is important for Tr1 development during parasitic and viral infections to prevent tissue damage, however they did not show the outcome of these infectious models where they need to show survival of mice after the infections or histology exhibiting tissue damage.

2. In Figure2, the author shows in vitro Tr1 differentiation by particular condition each. Since TCR signals from the environment are different based on their antigen, they need to compare different magnitude of TCR stimulation by which the impact of ITK signaling is consistent. It is known that insufficient stimulation of TCR induce T cell anergy. Although they showed Ki67 staining of CD4 T cells and IFN γ /IL-10 production, they need to clarify what is the alternative pathway for T cell differentiation without ITK signaling pathway to show specificity of ITK signaling cascade for Tr1 development.

3. In Figure3, The authors report AHR, c-Maf, and IRF4 expressions in Tr1 differentiation are affected by ITK signaling to support downstream molecular mechanism of Tr1 differentiation. These observations arise question whether the transcription factors for Th1 differentiation are induced in ITK signal deficient cells. To prove the specificity of ITK signaling for Tr1 differentiation, they need global comparison between WT and ITK signal deficient T cells. Furthermore, to emphasize the role of IRF4 under ITK signaling, they need to show what is the protein bypassing ITK and IRF4.

Minor points:

1. In Figure2d, the author showed IL-10 production from Tr1 cells is less by BMS treatment than CNX, however surface molecule expressions of Lag3, CD49b and PD-1 are less by CNX treatment than BMS. It is worthwhile to have a comment on this discrepancy.

2. In Figure2, the author showed IL-10 production from exTh17 cells was less in ITK signal deficient cells. To prove the effect is not due to cell survival, they should analyze the expression of transcription factors upregulated in Tr1 differentiation using these cells.

3. In Figure4, the author showed a suppression assay using differentiated Tr1 cells. To understand the function of Tr1 cells, it is better to include anti-IL-10 antibody treatment in the system to show how much of the suppressive effect is induced by IL-10.

In summary, this is an interesting paper showing that ITK signaling is important for Tr1 development. They also suggest a sequential interaction between ITK and IRF4 for IL-10 production from Tr1 cells. However, many of the key pieces of data are not provided to clearly support that this is the molecular mechanism responsible for the observed phenotypes.

Response to reviewer # 1's comments on NCOMMS-16-15371 "ITK signaling via IRF4 regulates the development and function of type 1 regulatory T cells". Changes are indicated by the highlighted areas in the revised manuscript:

1. Comment: "The induced-Tr1 cells cannot be called Tr1 cells without a full phenotyping and cytokine secretion profile being shown."

Response: Tr1 cells have been well characterized and were termed as Tr1 cells based on the previously described characterization by the groups of Flavell, Roncarolo, Kuchroo, Quintana, and others¹⁻⁴. This characterization is based on their expression of transcription factor Foxp3⁻ and cytokine IL-10⁺, and co-expression of surface markers LAG3 and CD49b. The purpose of this work was not to characterize Tr1 cells as a novel T cell subset, instead, to understand the role of ITK signaling in this previously well defined population.

2. Comment: "The levels of IL-10 production are only assessed at the protein level in humans, with one mouse experiment looking at mRNA levels, the authors should also assess changes in the levels of IL-10 mRNA transcripts in human cells, to determine if the inhibition of ITK is reducing IL-10 expression at both the transcriptional and translational level."

Response: We thank the reviewer for this suggestion, and now present data on the level of IL-10 transcripts in human CD4⁺ T cells polarized under Tr1-differentiating condition, in the presence or absence of ITK inhibitors (see **Fig. 3c**), (revision on pg. 7, line 138).

3. Comment: "As ITK is described to be important in TCR signaling is the phenotype of ITK^{-/-} cells a result of these cells not being as efficiently activated? Do the KO cells upregulate activation markers to the same extent? It is stated that they proliferated equally but data is not shown - this is important and should be included."

Response: This is a good question and one that we have tested. We conclude that this is not a result of the fact that the cells are not being activated, since all cells equally expressed Ki67, a marker of cells that are proliferating. We have now also added data of the T cell early activation markers CD25 and CD69 (**Fig. 2c**), and cell proliferation dye dilution (**Fig. 2d**), suggesting that at a minimum, the cells are able to receive sufficient signal to cause them to become activated and proliferate in the absence of ITK (revision on pg. 5, line 108).

4. Comment: "At the first introduction of ITK it should be named in full and discussed in more detail, a comment in the introduction on known differences in downstream signals of TCR/ITK would help contextualise the data. It should also be briefly described how efficient the ITK KO model is i.e. what is the percent residual kinase activity?"

Response: This was an oversight on our part since the *Itk*^{-/-} mice have been so well characterized and described in the literature. We have now included the full name of ITK and a statement on the previously described phenotypes of the *Itk*^{-/-} mouse model in the introduction. Note that the *Itk*^{-/-} mouse model is a complete knockout of the *Itk* gene, and no protein is expression, and so there is no residual kinase activity (revised on pg. 3, line 57, 65).

5. Comment: “Only a few gating strategies are shown for isolation of starting cell populations; these should be included as supplemental data for all FACS analysis/sorting approaches. It is also unclear whether viability staining was performed in all experiments and whether dead/dying cells have been excluded from analyses/sorting.”

Response: We have now included gating strategies in the supplementary materials, including gating for naïve CD4⁺ T cell sorting, cultured and freshly isolated T cells, retroviral transfected T cells, and T cells in suppression assays (**Supplemental Fig. 7**). A fixable viability dye staining was performed in all analyses as addressed in the original manuscript, METHOD/Flow cytometry section.

6. Comment: “In the methods it is stated that for in vitro differentiation of mouse Tr1 cells there was varying amounts of IL-27 (20-25ng/mL), anti-IFN γ , anti-IL-12 (5-10ug/mL) added and TGFbeta was sometimes added. Also that analysis was performed 48-72h later. These are large variations and without knowing exactly which culture conditions/time of induction was used in each experiment it is hard to compare the data, particularly as cytokine profiles can vary greatly from 48-72h - the method used to generate data in each experiment needs to be clear. Why was this time (48-72h) chosen as a read out? Does the IL-10 profile change over longer period of in vitro culture/with repeated TCR stimulation?”

Response: We have added a supplementary figure indicating that TGF- β has limited effect in Tr1 cell differentiation *in vitro* in our hands (**Supplementary Fig. 4**), and so we used data generated without TGF- β in the main figures of the manuscript. We have also included the kinetics of Tr1 cell differentiation *in vitro*, which showed that percentage of IL-10⁺Foxp3⁻ Tr1 cells differentiated from WT and *Itk*^{-/-} naïve CD4⁺ cells exhibited significant difference from 48 hours on and remained significantly different even up to 144 hours from the initial culture time point (**Fig. 1**) (revised on pg. 5, line 105).

7. Comment: “The culture conditions for mice Tr1 cells is quite different to that used for human Tr1 cell differentiation where IL-2, IL-15 and IL-10 are added. It has been shown that some *Itk*^{-/-} effects on Th polarization is due to altering IL-2 sensitivity, does adding these additional cytokines to mouse cultures affect the results?”

Response: We thank the reviewer for this interesting suggestion. We have tested the effect of IL-2 in mouse Tr1 cell differentiation. Unlike the full rescue, or exactly speaking, over induction of *Itk*^{-/-} Th9 differentiation by IL-2⁵, we only observed a moderately partial rescue of *Itk*^{-/-} Tr1 cell differentiation by high dose of IL-2 (10 ng/ml, roughly 2000 U/ml). This data is now included in **Supplementary Fig. 4**. We have also added a discussion, with comparison of this observation to that reported in ITK function during Th9 differentiation⁵ (pg. 11, line 229).

8. Comment: "Fig 1A. The phenotype of the generated dual reporter *ITK*^{-/-} should be briefly described, do they develop any spontaneous autoimmunity, have defects on any other cell types. Do they have the expected phenotype of increased Th1 and Tregs and decreased Th17, Th2 and Th9? The bar graphs showing %Tr1 - what is this a percent of? How are induced Tr1 cells defined, seems to be only by IL-10 expression, without a more extensive phenotype shown of the IL-10⁺ cells this is insufficient. Given this is such a drastic phenotypic difference it would be good to also show by direct protein staining and at mRNA level for IL-10 (and IL-4 as true Tr1 cells should be IL-4neg).

Response: As discussed above in reference to the previous query, the *Itk*^{-/-} mouse model has been well described and they have not been reported to develop spontaneous autoimmune disease. The crosses that generate the IL-10^{GFP} and Foxp3^{RFP} dual reporter mice have been extensively explored⁶. Additions of the reporters do not generate any additional modifications to the IL-10 or Foxp3 genes, but merely report on the expression of these genes. So these mice are still only *Itk* deficient and so behave the same as the *Itk*^{-/-} that are not crossed to the reporter mice. During our investigation, in addition to Tr1 cell differentiation, we have used the WT and *Itk*^{-/-} IL-10^{GFP}/Foxp3^{RFP} dual reporting cells for Th1, Th17 and iTreg cell differentiation (**Reviewer Fig. 1, a**), with results consistent with what we and our colleagues reported previously⁷⁻¹⁰. The bar graphs showing % Tr1 are a percentage of IL-10⁺Foxp3⁻ CD4⁺ T cells among viable CD4⁺ T cells. We have clarified this in the legends. The characterization of whether they are Tr1 cells has been done as described in the other figures and discussed above in response to the previous query, i.e. profile of expression of surface marker (CD49b/LAG3), transcription factor (AHR, cMAF, IRF4) and most importantly, IL-10/Foxp3. IL-4 is very low or not detectable in cells with this combination phenotype³. Consistently to what others found, we have observed low level of IL-4 expression in CD4⁺ T cells cultured using our Tr-1 polarizing condition (**Reviewer Fig. 1, b**).

Reviewer Figure 1. (a) Naïve CD4⁺ T cells isolated from WT and *Itk*^{-/-} IL-10^{GFP}/Foxp3^{RFP} dual reporter mice were cultured under Th1, Th17, iTreg, and Tr1 cell differentiating condition. Representative plots showing IL-10/IFN- γ expression (by intracellular cytokine staining) in Th1-polarized, IL-10/IL-17A expression (by intracellular cytokine staining) in Th17-polarized, and IL-10^{GFP}/Foxp3^{RFP} expression (direct reporting by GFP/RFP) in iTreg and Tr1-polarized CD4⁺ cells. (b) Representative plots of IL-10/IL-4 expression (by intracellular cytokine staining) in CD4⁺ T cells cultured under Tr1-polarizing or Th2-polarizing conditions.

9. Comment: “Fig 1B. The authors focus on the ability to induce Tr1 cells from the ITK KO mice, although it appears that there is no difference in the levels of circulating in vivo derived IL-10⁺ CD4⁺ T cells, this is not discussed (levels in spleen actually look higher in KO mice!). It would be interesting to see if the same phenotype of reduced cMAF, AhR and IRF4 is evident in the IL-10⁺ cells isolated from KO mice vs WT without any manipulation. Title '% of Tr1' appears to be a typo.”

Response: The IL-10⁺ T cells in *Itk*^{-/-} spleen in the steady state exhibit low level of LAG3 and CD49b expression, so it is unlikely that these are Tr1 committed cells. Given the low number of these cells, our attempt to isolate and fully characterize these cells with our available mouse stock has not been successful. We have modified the figure legend to clarify this and added the LAG3/CD49b staining data, along with our discussion about this phenotype in the steady state. We thank the reviewer for pointing this out and hope that this clarification is acceptable (revised on pg. 5, line 92, 113).

10. Comment: “Fig 1C. I am glad to see these data showing pathogen-driven in vivo generation of IL-10 producing cells, this is important, although again a full phenotype should be shown to call them Tr1 cells. Why were the lungs isolated for analysis of CD4⁺ T cells following helminth infection? Are the same differences seen in the MLN, peripheral blood, spleen etc as per 1B?”

Response: We have added the data of LAG3/CD49b expression to verify the Tr1 cell phenotype (new **Fig. 1d**). *N.B.* travel through the lungs to mature, until about 5 days post infection, before homing to mouse guts where they are expelled¹¹. In our hands, 7 days post *N.B.* infection, CD4⁺ T cells isolated from the mesenteric lymph nodes and spleen showed moderate level of induction of IL-10 production and small or no increase in number of Tr1 cells (**Supplementary Fig. 6g**). Cells

isolated from the lungs of infected mice showed the most significant induction of IL-10⁺Foxp3⁻LAG3⁺CD49b⁺ Tr1 cells (**Fig. 1c & d**).

11. Comment: “Fig 1D. Interestingly the viral infection is driving more IL-10 induction than the helminth infection in 1C, perhaps the authors could comment on this and also that there doesn't appear to be a significant affect on the ability of FOXP3+ cells to produce IL-10 in the ITK^{-/-} mice. Is this defect in IL-10 induction restricted to the lung or generally seen throughout the mice?”

Response: Influenza A viral infection has been reported to trigger high levels of IL-10 production in T cells in the lungs, moderate levels of IL-10 expression in T cells in the pulmonary draining lymph nodes, but very limited induction of IL-10 expression in the spleen¹². We have seen similar trends, and added the data that Tr1 cells were higher in WT draining lymph nodes than in *Itk*^{-/-} mice, and that there is no induction and so no difference in the spleens (**Supplementary Fig. 6h**). In this work, Foxp3 is mainly used as a marker in Tr1 cell definition and to exclude the classical Foxp3⁺ regulatory T cells from our gating of the IL-10⁺ Tr1 cells. The findings on the Foxp3⁺ regulatory T cells are continuing to be investigated and are beyond the scope of this report.

12. Comment: “Fig 2A. Statistical differences between *Itk*^{-/-} and *Itk*^{+/+}3MBPP1 should be shown for all graphs.”

Response: We have modified legends as the reviewer suggested.

13. Comment: “Fig 2B. The expression levels of the Tr1 markers are only shown separately, what is the percent of LAG3+CD49b+ cells within IL-10+ cells? Dot plots should be shown alongside bar graphs and must include the gating strategy showing how positive gates were set.”

Response: Using gating shown in **Fig. 2e**, we have determined the percentage of LAG3⁺CD49b⁺ cells within IL-10⁺ Tr1 cells, and changed the MFI data into percentage data as suggested by the reviewer.

14. Comment: “Fig 2C. Is the same effect seen with TCR re-stimulation as opposed to PMA/Ionomycin stimulation?”

Response: ITK is downstream of the TCR, and it is not surprising that when stimulated through TCR by anti-CD3/CD28, we were unable to induce rapid IL-10 production that can be captured within hours by the Brefeldin A and Monensin. Given the other major data on Tr1 cell development added to this revised manuscript, we felt that it is better to relocate the data on the function of ITK in regulating the balance IL-10/IFN- γ production into the supplementary information and engage it during discussion, so as to maintain the focus on the work on its role in Tr1 differentiation and function.

15. Comment: “Fig 2D. Same comment as for mice that full Tr1 phenotype need to be shown in human cells. Fig 2E. Dot plots showing gating are required and an analysis of %LAG3+CD49b+ cells within IL-10+ cells. Suppression assays should be performed with human induced ‘Tr1 cells’ to confirm a regulatory phenotype.”

Response: We have added the dot plots for viability, proliferation and LAG3/CD49b gating. Given that in ITK deficient and inhibited cells, IL-10⁺ cell counts were extremely low, and that LAG3⁺CD49b⁺ identifies the Tr1 cells from total CD4⁺ T cells³, we have plotted LAG3/CD49b using total viable CD4⁺ cells (**Fig. 3 b & d**). The LAG3/CD49b double positive Tr1 cells have been extensively investigated in their suppressive function in both human and mouse³. Note that we are unable to evaluate the role of ITK in the suppressive ability of the human Tr1 cells since, in the co-culture system, inhibiting ITK in the human Tr1 cells also inhibits ITK activity in the responding cells. However, we are able to do this experiment only because we have the Tr1 cells from the ITKas expressing mice, which we can uniquely inhibit ITK activity of the *Itkas* Tr1 cells with the 3MBPP1, without affect the ITK activity in the responding cells (**Fig. 4**).

16. Comment: “Fig 2F. A better demonstration of trans-differentiation would be to isolate *in vivo* differentiated Th17 cells and culture in Tr1 conditions as the induced Th17 probably do not represent a truly committed Th17 cell population. What was the amount of IL-10 (and other cytokines/Tr1 markers) expression in cells in Th17 condition prior to sorting and culture in Tr1 condition? Do they also upregulate Tr1 markers along with IL-10 expression following culture in Tr1 conditions. If the IL17/FOXP3 reporter mice are infected as per Fig 1 do they also have reduced trans-differentiation *in vivo*? This should also be shown using *ex vivo* isolated human Th17 cells. Currently there is insufficient data in this figure to conclude that ITK affects trans-differentiation of committed Th17 cells into Tr1 cells.”

Response: To evaluate the Th17 trans-differentiation in live CD4⁺ T cells *in vivo*, a more sophisticated transgenic mouse model that allow marking of the history of IL-17A production and the real-time reporting of IL-17A/IL-10/Foxp3 expression is required. For example, Flavell and colleagues crossed [(IL-17A^{CRE}) × (Rosa26 STOP^{fl/fl} YFP)] with [(IL-17A^{Katushka}) (IL-10^{GFP}) (Foxp3^{RFP})] reporter mouse models¹³. This is a quintuple (5; each **bracket** indicates one transgene) transgenic modification, and to generate *Itk*^{-/-}, it becomes sextuple (6), furthermore to generate *Itkas*-KI using *Itk*^{-/-}, it is eventually a septuple (7) transgenic mouse-breeding program. Due to limited time, funds and space, we are unable to do this. Given the other major data on Tr1 cell development added to this revised manuscript and the lack of *in vivo* data to support the function of ITK in Th17 trans-differentiation in Tr1 cells, we felt that it is better to relocate the data of the possible function of ITK in regulating the exTh17 cell trans-differentiation into Tr1 cells into the supplementary information and engage it with the current manuscript in discussion. We have also added data about viability,

proliferation, and transcription factors in the supplementary information for discussion (**Supplementary Fig. 5**).

17. Comment: “Fig 3. It is unclear what 'background' histogram represents. Again, dot plots showing gates are required for all data only shown in histogram or bar plots. Are the differences in cMAF due to differences in culture conditions between mice and humans?”

Response: The “background” level was the naïve CD4⁺ T cell level used as control. We have added the dot plots and changed the legends as suggested by the reviewer. It is possible that the discrepancy in cMAF expression is due to the difference in culture conditions between mouse and human cells, and we have added this discussion (revised on pg. 8, line 166). We thank the reviewer for this helpful comment.

18. Comment: “Fig 3G. It is stated that IRF4 re-expression rescued Tr1 cell development, this appears to only be a partial rescue and text should be amended to reflect percent rescue and again is entire phenotype rescued or only IL-10 expression? Show stats between *Itk*^{-/-} and WT populations on bar graph.”

Response: It is unlikely that we would get complete rescue given the fact that the reintroduced IRF4 is driven by a non-native promoter and need to be incorporated after cells are activated overnight. However, our data suggest that IRF4 expression is a critical factor regulated by ITK that explains the defect seen in the absence of *Itk*. We have added data showing that the other Tr1 markers LAG3 and CD49b are also rescued. As suggested by the reviewer, we have amended the text to discuss this partial effect and included data on the other markers of Tr1 cells, as well as the statistical analysis of *Itk*^{-/-} and WT cells in the bar graphs (revised on pg. 9, line 179, 189).

19. Comment: “This finding for IRF4 seems to be a key piece in the Tr1 puzzle. Do the authors have data on the relative expression of IRF4 in WT 'Tr1 cells' compared to Tregs and other Th cells to show that this is a feature of Tr1 cells? It is disappointing that suppression assays were not done with the IRF4 transduced cells to see if IRF4 re-expression can also rescue suppressive function.”

Response: We have previously reported that ITK is required for Th17 differentiation but IRF4 expression in *Itk*^{-/-} CD4⁺ T cells cultured under Th17-differentiating condition is normal⁷. The role of IRF4 downstream of ITK in T cell development and function is a long# term interest of our group and we continue to explore this, however, this work is beyond the scope of the current manuscript.

[Redacted]

We have also isolated WT and *Itk*^{-/-} IL⁺ 10⁺ Foxp3⁺ Tr1 cells that are IRF4⁺ RV⁺ or IRF4⁺ RV⁻ and performed the Tr1 suppression assay and found that IRF4 re⁺ expression rescued the *Itk*^{-/-} Tr1 cell suppressive function (**Fig. 6c**).

[Redacted]

20. Comment: “Fig 4. Gating needs to be shown for assay analysis, in particular were dead responders excluded from analysis and was %death different between conditions. Rather than % proliferation % suppression should be shown, calculated using the division index as per McMurchy and Levings 2012 EJI. It is insufficient to only show suppression assays at a 1:1 ratio, a range of suppressor: responder cells to at least a 1:16 ratio should be shown. It would have been nice to see a direct investigation of the importance of IL-10 secretion for suppressive function through either transwell assays, adding IL-10 bAb or IL-10R nAb or by adding back recombinant IL-10 to the *Itk*^{-/-}+3MBPP1 wells to same levels seen in WT Tr1 well to see if suppression could be restored.”

Response: As discussed in the response to the gating query above, all flow cytometry experiments include gating for singlets and for viability, including all Tr1 suppression assays (**Supplementary Fig. 7e**). We have expressed the data as % suppression as suggested by the reviewer. In brief, Division Index (DI) was calculated using the FlowJo “Proliferation Platform”, in which Division Index (DI)

is defined as the ratio of “the total number of divisions/the number of cells at start of culture”. Then the average of DI of responder cells with anti-CD3 stimulation but no Tr1 cell suppression was named as DI_0 , while the DI of responder cells under condition i was named as DI_i . The percentage of suppression (PS) under condition i was defined as $PS_i = 100 \cdot (1 - DI_i / DI_0)^{14}$ (revised on pg. 19, line 413).

21. Comment: “It should be made clear in the discussion that the role for Itk in Tr1 development has mainly been shown in vitro and it is unexplored whether it is required for in vivo differentiation in humans.”

Response: We have updated the discussed on this point as suggested by the reviewer (revised on pg. 14, line 306).

22. Comment: “The PCR cycling conditions and machine used should be provided.”

Response: We have updated the materials and methods section on this point as suggested by the reviewer (revised on pg. 21, line 454).

23. Comment: “In some figure legends the number of mice/patients used for each experiment and the number of experiments that are pooled are unclear - fix throughout. It is not enough to state $n > x$, the exact n should be stated.”

Response: We have updated the legends to the relevant figures on these points as suggested by the reviewer, throughout the manuscript.

24. Comment: “Most statistical analysis used Student's t tests were used, was a test of normality done to determine normal distribution of the data sets being analysed? Otherwise the non-parametric Mann-Whitney test is a more suitable test

Response: We have changed to the non-parametric Mann-Whitney test for all single group comparisons.

We thank the reviewer for the detailed comments and helpful suggestions, and hope that these changes bring the manuscript to the level where it can be accepted for publication.

Response to reviewer # 2's comments on NCOMMS-16-15371 "ITK signaling via IRF4 regulates the development and function of type 1 regulatory T cells". Changes are indicated by the highlighted areas in the revised manuscript:

1. Comment: "The data presented in the paper are clear, that ITK signaling under TCR stimulation is important for Tr1 cell development and function, however the importance of TCR signaling for T cell differentiation is expected and therefore detract somewhat from novelty of the observations unless they show specificity of this cascade compared to other TCR signaling cascades."

Response: Given the common pathways used by the TCR for T cell differentiation to different CD4 effector cell types, it would indeed be a surprise that they do not share such pathways. However, as we discussed in the introduction, ITK pathways are not required for the development of Foxp3⁺ regulatory T cells, providing specificity for ITK in the generation of different regulatory T cell lineages, suggesting some specificity as sought by the reviewer.

[Redacted]

2. Comment: "Furthermore, they did not show which protein is phosphorylated by ITK and binds on IRF4 transcription promoting region. Current analysis is not sufficient for proving this molecular mechanism of TCR/ITK/IRF4 pathway."

Response: Our conclusion that there is a TCR/ITK/IRF4 pathway is based on the reduced expression of IRF4 in the absence of ITK, and the rescue of differentiation when IRF4 is re-expressed. We do not suggest that this is a direct pathway, i.e. that ITK phosphorylates IRF4 or some upstream factor that regulates IRF4, but that there are a pathway(s) regulated by ITK kinase activity which leads to IRF4 expression. We have updated the manuscript with our recent finding that the RAS/MAPK signaling activation positively regulates Tr1 cell differentiation, and that constitutively active HRas expression fully rescued *Itk*^{-/-} CD4⁺ Tr1 cell differentiation to the WT levels, along with a rescue of IRF4 expression (**Fig. 8**). We have revised the title accordingly as well. We hope that this additional finding raise this work to level acceptable as sought by the reviewer.

3. Comment: "In Figure1, the authors are claiming ITK signaling is important for Tr1 development during parasitic and viral infections to prevent tissue damage, however they did not show the outcome of these infectious models where they need to show survival of mice after the infections or histology exhibiting tissue damage."

Response: We have included the parasite burden (*N.B.* model), weight curve, survival curve, viral PFU (WSN model), and histology of lung tissues (both

models) in the supplementary information (**Supplementary Fig. 6**). We have seen higher worm burden in *N.B.*-infected and high death rate in WSN-infected *Itk*^{-/-} mice than WT controls. Given the complex role of ITK in anti-parasitic Th2 and antiviral Tc1 cell development, it is hard to attribute all phenotypes to its function in Tr1 cells. We used these infectious disease models to trigger Tr1 cell development *in vivo*, with the aim of determining ITK function in Tr1 cell differentiation *in vivo*. We had discussed these outcomes accordingly (revised on pg. 13, line 284).

4. Comment: “In Figure2, the author shows *in vitro* Tr1 differentiation by particular condition each. Since TCR signals from the environment are different based on their antigen, they need to compare different magnitude of TCR stimulation by which the impact of ITK signaling is consistent. It is known that insufficient stimulation of TCR induce T cell anergy. Although they showed Ki67 staining of CD4 T cells and IFN γ /IL-10 production, they need to clarify what is the alternative pathway for T cell differentiation without ITK signaling pathway to show specificity of ITK signaling cascade for Tr1 development.”

Response: Defective IL-10⁺ Foxp3⁻ Tr1 cell differentiation has been observed *in vitro* under Tr1 polarizing condition, as well as *in vivo* in tissues that are subjected to parasitic or viral exposure. We now also show in **Fig. 2** that early activation of WT and *Itk*^{-/-} cells under Tr1 polarizing conditions are both sufficient for T cell division, while the Tr1 related markers including IL# 10, LAG3, CD49b, ICOS, and PD# 1 are severely impaired in the absence of ITK.

[Redacted]

5. Comment: “In Figure3, The authors report AHR, c-Maf, and IRF4 expressions in Tr1 differentiation are affected by ITK signaling to support downstream molecular mechanism of Tr1 differentiation. These observations arise question whether the transcription factors for Th1 differentiation are induced in ITK signal deficient cells. To prove the specificity of ITK signaling for Tr1 differentiation, they need global comparison between WT and ITK signal deficient T cells. Furthermore, to emphasize the role of IRF4 under ITK signaling, they need to show what is the protein bypassing ITK and IRF4.”

Response: It is not clear what the reviewer means with the statement: “Furthermore, to emphasize the role of IRF4 under ITK signaling, they need to show what is the protein bypassing ITK and IRF4”. In *Itk*^{-/-} CD4⁺ T cells, IRF4 expression under the Tr1 polarizing condition was impaired as well as the expression of IL-10. We have shown in this revised manuscript that the activation of HRas signaling rescued the IRF4 and IL# 10 expression in *Itk*^{#/#} cells to the level that observed in the normal WT cells (**Fig. 8**).

[Redacted]

6. Comment: “In Figure2d, the author showed IL-10 production from Tr1 cells is less by BMS treatment than CNX, however surface molecule expressions of Lag3, CD49b and PD-1 are less by CNX treatment than BMS. It is worthwhile to have a comment on this discrepancy.”

Response: We utilized these inhibitors of ITK on human T cells to show that 2 different ITK inhibitors are able to affect this pathway in these cells. However, these two inhibitors have different characteristics and IC_{50} s, as well as off target effects, and so it is likely that the difference between these two inhibitors is a reflection of this. Note that our work with the 3MBPP1 is much more indicative of what would happen when we only inhibited ITK activity, however, we are not able to use this inhibitors in the human T cells for obvious reasons. We have discussed this in the revised discussion (revised pg. 13, line 279).

7. Comment: “In Figure2, the author showed IL-10 production from exTh17 cells was less in ITK signal deficient cells. To prove the effect is not due to cell survival, they should analyze the expression of transcription factors upregulated in Tr1 differentiation using these cells.”

Response: Given the new major data on Tr1 cell development added to this revised manuscript and the lack of *in vivo* data to support the function of ITK in Th17 trans-differentiation in Tr1 cells, we felt that it is better to relocate the data on the potential function of ITK in regulating the exTh17 cell trans-differentiation into Tr1 cells into the supplementary information and engage it with the current manuscript in discussion. We have also added data about viability, proliferation, and transcription factors in the supplementary information for discussion.

8. Comment: “In Figure4, the author showed a suppression assay using differentiated Tr1 cells. To understand the function of Tr1 cells, it is better to include anti-IL-10 antibody treatment in the system to show how much of the suppressive effect is induced by IL-10.”

Response: It has been recently shown by Gagliani, Roncarolo and colleagues that the $IL-10^+$ $LAG3^+$ $CD49b^+$ Tr1 cells suppress responder T cell proliferation in an IL-10-dependent manner³. This data was in the supplementary information of their

Supplementary Fig. 7d from Gagliani *et al.*, [doi:10.1038/nm.3179](https://doi.org/10.1038/nm.3179)

published article, so may not be easy to find. We have attached it here for the reviewer's interest (**Supplementary Fig. 7d** from Gagliani *et. al.*³).

We thank the reviewer for the insightful comments and hope that these changes bring the manuscript to the level where it can be accepted for publication.

References:

1. Mascanfroni, I.D. *et al.* Metabolic control of type 1 regulatory T cell differentiation by AHR and HIF1- α . *Nat Med* **21**, 638-646 (2015).
2. Apetoh, L. *et al.* The aryl hydrocarbon receptor interacts with c-Maf to promote the differentiation of type 1 regulatory T cells induced by IL-27. *Nat Immunol* **11**, 854-861 (2010).
3. Gagliani, N. *et al.* Coexpression of CD49b and LAG-3 identifies human and mouse T regulatory type 1 cells. *Nat Med* **19**, 739-746 (2013).
4. Brockmann, L. *et al.* IL-10 Receptor Signaling Is Essential for TR1 Cell Function In Vivo. *J Immunol* (2016).
5. Gomez-Rodriguez, J. *et al.* Itk is required for Th9 differentiation via TCR-mediated induction of IL-2 and IRF4. *Nat Commun* **7**, 10857 (2016).
6. Huber, S. *et al.* Th17 cells express interleukin-10 receptor and are controlled by Foxp3(-) and Foxp3+ regulatory CD4+ T cells in an interleukin-10-dependent manner. *Immunity* **34**, 554-565 (2011).
7. Gomez-Rodriguez, J. *et al.* Differential expression of interleukin-17A and -17F is coupled to T cell receptor signaling via inducible T cell kinase. *Immunity* **31**, 587-597 (2009).
8. Gomez-Rodriguez, J. *et al.* Itk-mediated integration of T cell receptor and cytokine signaling regulates the balance between Th17 and regulatory T cells. *J Exp Med* **211**, 529-543 (2014).
9. Huang, W., Jeong, A.R., Kannan, A.K., Huang, L. & August, A. IL-2-inducible T cell kinase tunes T regulatory cell development and is required for suppressive function. *J Immunol* **193**, 2267-2272 (2014).
10. Kannan, A.K., Sahu, N., Mohanan, S., Mohinta, S. & August, A. IL-2-inducible T-cell kinase modulates TH2-mediated allergic airway inflammation by suppressing IFN- γ in naive CD4+ T cells. *J Allergy Clin Immunol* **132**, 811-820 e811-815 (2013).
11. Camberis, M., Le Gros, G. & Urban, J., Jr. Animal model of *Nippostrongylus brasiliensis* and *Heligmosomoides polygyrus*. *Curr Protoc Immunol* **Chapter 19**, Unit 19 12 (2003).
12. Sun, J., Madan, R., Karp, C.L. & Braciale, T.J. Effector T cells control lung inflammation during acute influenza virus infection by producing IL-10. *Nat Med* **15**, 277-284 (2009).
13. Gagliani, N. *et al.* Th17 cells transdifferentiate into regulatory T cells during resolution of inflammation. *Nature* **523**, 221-225 (2015).
14. McMurchy, A.N. & Levings, M.K. Suppression assays with human T regulatory cells: A technical guide. *Eur J Immunol* **42**, 27-34 (2012).

REVIEWERS' COMMENTS:

Reviewer #1 (Remarks to the Author):

Review of: 'ITK signaling via the Ras/IRF4 pathway regulates the development and function of type 1 regulatory T cells'

Summary: This revised manuscript has been substantially improved and includes new data that expand upon the previously described role for ITK signaling in the differentiation and function of Tr1 cells. The new data focus on the factors downstream of ITK that are critical for the observed effects on Tr1 cells and eliminate Blimp-1 while implicating a requirement for the Ras/IRF4 pathway.

General comment: My primary concern from the original manuscript, that remains the same with the revised manuscript, is the appropriate application of the term 'Tr1 cells' without showing in the specific models and culture systems used the complete Tr1 phenotype. This is not helped by a lack of consensus in the literature on the definition for Tr1 cells in both mice and humans. The definition most commonly used is suppressive FOXP3-IL-10+IL-4neg cells with intermediate IFN γ (compared to Th1 cells). More recently has been the addition of co-expression of LAG-3 and CD49b; and while this seems fairly robust in mice, there is conjecture on how specific this is for humans (<https://doi.org/10.3389/fimmu.2016.00355>). This refers back to my previous comments from the first review where I asked that the full phenotype of the Tr1 cells be shown at the first instance, regardless of if they have been previously described, this study used different models and methods of induction and therefore a full phenotype is needed to call them Tr1 cells. Thankyou for providing the data in reviewer Fig 1b and this would be helpful to partially address this comment by adding to the supplemental material.

It needs to be clearly stated at the beginning of the results section what is being defined and termed a 'Tr1 cell' in mice and human studies in this paper. It is crucial to make this clear early on as <50% of the CD4+FOXP3-IL-10+ are LAG3+CD49b+ in WT mice and <15% in the *Itk*^{-/-} mice. Particularly for the human data terminology such as 'Tr1-like' or stating the phenotype (e.g. IL-10+FOXP3- T cells) would be more appropriate. Indeed, the title would be more accurate to replace Tr1 cells with CD4+ FOXP3-IL-10+ T cells.

The authors have satisfactorily addressed the majority of my comments from the first review and, aside from the general comment above, the following specific comments are directed against any remaining areas requiring further clarification and new data. [Redacted]

Specific Comments:

Fig 2c and d. These important new data that show no deficit in activation or proliferation in *ITK*^{-/-} cells would benefit from having a bar graph similar to e and f showing the collated data alongside the dotplots.

Fig 3. The additional data showing viability, Ki67 and co-expression of LAG3 and CD49b in the cultured cells supports the original claims regarding inhibition of ITK function. However, as per Fig 1 it would be informative to see the percent of LAG3+CD49b+ cells within the CD4+FOXP3-IL-10+ cell population or vice versa.

Fig 7. Expression levels of Blimp-1 after treatment with the different vectors need to be shown as per IRF4 in Supp Fig 3a.

Supp Fig 2b. It is unclear after the 60h of culture under Tr1-conditions how long the cells were stimulated with anti-CD3/28 in presence of Bref A and Monensin. Can the authors also state what

number of events was used as a cutoff, below which the value was adjusted to zero?

Supp Fig 3. The addition of symbols for significant difference from naïve cells is not particularly useful and makes the figure more cluttered, could state in legend that all culture conditions had significantly increase 'xx' compared to naïve T cells. Similarly, it is not particularly informative to show differences between grouped conditions. It is important to see differences between WT and ITK^{-/-} in both control-RV and IRF4-RV groups (as already shown) as well as between WT and ITK^{-/-} cells from each group.

In the new results section on the role of HRas (lines 211-215) it is stated that expression of Hras can rescue Tr1 cell development...which is further enhanced by the expression of constitutively active HRas mutant. Data in Fig 8d show that the HRas-RV in ITK^{-/-} cells can partially rescue IL-10⁺ cell development (compared to WT) while the constitutively active HRasG12V-RV can fully rescue the IL-10⁺ cells.

Minor Comments:

Page 5, line 107 should refer to figure 2 not figure 1.

Page 12 lines 246-248 the sentence 'Our data suggest that re-expression of IRF4 also rescues the function of the Tr1 cells that developed suggest that this factor is sufficient, to complement ITK signals for the differentiation and function of Tr1 cells' does not make sense. Perhaps it should read more along the lines of 'the data suggest that IRF4 is sufficient to compensate for a lack of ITK as re-expression of this IRF4 rescues 'Tr1 cell' differentiation and function.

Page 14, line 302 the phrase 'may be a promising strategy in unleashing the immune depression' does not make sense.

In methods section on Human Tr1 differentiation in vitro page 17 line 375 please describe what 'full RPMI-1640 medium' contains.

Supp Fig 6 panel f is referred to as (b).

All figure legends now state an 'n', although some ambiguity is still present. As one example Fig 2 states n=6. Data represent results from more than three independent experiments. So data in all panels of this figure are pooled from n=6 mice from >3 experiments? Need to be more specific and state 'data are from between 3-x independent experiments'. This should be fixed throughout all legends.

Reviewer #2 (Remarks to the Author):

The authors have addressed all the concern that I had raised in the initial review, in some cases with addition of new data.

Response to reviewer # 1's comments on NCOMMS-16-15371A "ITK signaling via IRF4 regulates the development and function of type 1 regulatory T cells". Changes are indicated by the highlighted areas in the revised manuscript:

1. Comment: "My primary concern from the original manuscript, that remains the same with the revised manuscript, is the appropriate application of the term 'Tr1 cells' without showing in the specific models and culture systems used the complete Tr1 phenotype. This is not helped by a lack of consensus in the literature on the definition for Tr1 cells in both mice and humans. The definition most commonly used is suppressive FOXP3-IL-10+IL-4neg cells with intermediate IFN γ (compared to Th1 cells). More recently has been the addition of co-expression of LAG-3 and CD49b; and while this seems fairly robust in mice, there is conjecture on how specific this is for humans (<https://doi.org/10.3389/fimmu.2016.00355>). This refers back to my previous comments from the first review where I asked that the full phenotype of the Tr1 cells be shown at the first instance, regardless of if they have been previously described, this study used different models and methods of induction and therefore a full phenotype is needed to call them Tr1 cells. Thank you for providing the data in reviewer Fig 1b and this would be helpful to partially address this comment by adding to the supplemental material."

Response: As we are sure that the reviewer has appreciated, unlike Foxp3⁺ T regulatory cells, where there is clear consensus on the definition and markers that identify those cells, most of the published work in this area has defined the Tr1 cells as Foxp3⁻ IL-10-producing CD4⁺ T cells, and this is the definition that we agreed on. The work by Gagliani and colleagues¹ attempted to identify markers that would allow analysis of these cells without having to fix and permeabilize them (to analyze IL-10 and Foxp3 status) so that they can be analyzed for potential clinical applications, leading to the identification of the cell surface markers LAG3 and CD49b. These additional markers are used in our work to further characterize the phenotype of the Tr1 cells induced and align our work with others in the field. In the recommended reference by the reviewer, Tr1 cells were also defined as CD4⁺ Foxp3⁻ IL-10⁺ cells (while IL-10-producing CD4⁺ T cells were termed as bulk Tr1-like cells, when Foxp3 status was undetermined). However, this work raised concern that human IL-10-producing CD4⁺ T (Tr1-like, Foxp3 status undetermined) cells do not always express LAG3 and CD49b. The discrepancy of LAG3/CD49b expression in these IL-10-producing human CD4⁺ T cells may due to the difference of the initial stages of the cells used and/or the conditions used to differentiate IL-10-producing cells. We have included discussion relevant to these observations on page 13, line 273. In accordance with the reviewer's request, we have included the data we provided in the previous Reviewer Fig. 1b to the revised supplemental material as the new **Supplementary Fig. 7**.

2. Comment: "It needs to be clearly stated at the beginning of the results section what is being defined and termed a 'Tr1 cell' in mice and human studies in this

paper. It is crucial to make this clear early on as <50% of the CD4+FOXP3-IL-10+ are LAG3+CD49b+ in WT mice and <15% in the *Itk*^{-/-} mice. Particularly for the human data terminology such as 'Tr1-like' or stating the phenotype (e.g. IL-10+FOXP3- T cells) would be more appropriate. Indeed, the title would be more accurate to replace Tr1 cells with CD4+ FOXP3-IL-10+ T cells."

Response: As the reviewer suggested, we had clearly defined the term "Tr1 cells" as those expressing "high levels of IL-10 but no Foxp3" in the original abstract and introduction, and have also now added a line of clarification in the result section in page 5, line 83 that we are investigating the role of ITK in "CD4⁺ Foxp3⁻IL-10⁺ Tr1 cell development". With regards to the title we believe that the current title is appropriate given the description we and others have presented for these cells.

3. [Redacted]

4. Comment: "Fig 2c and d. These important new data that show no deficit in activation or proliferation in *ITK*^{-/-} cells would benefit from having a bar graph similar to e and f showing the collated data alongside the dot plots."

Response: Due to space limit, the means and standard error means of **Fig. 2c** and **Fig. 2d** (now **Fig. 3c & d**) are included in the dot plot figure annotations, and asterisks below time points indicate the significance of difference. These were stated in the original figure legends and we have added another line of clarification to specify this in the legends of the new **Fig. 3c & d**. We hope that it is now clear to the reviewer.

5. Comment: "Fig 3. The additional data showing viability, Ki67 and co-expression of LAG3 and CD49b in the cultured cells supports the original claims regarding inhibition of *ITK* function. However, as per Fig 1 it would be informative to see the percent of LAG3+CD49b+ cells within the CD4+FOXP3-IL-10+ cell population or vice versa."

Response: As we described in the method section, the antibodies we used for human FOXP3 (APC) and LAG3 (Alexa Fluor 647) were in the same fluorescent channel, precluding us from examining these 2 markers together. We therefore had to perform the LAG3/CD49b staining and FOXP3/IL-10 staining separately when characterizing the human Tr1 cells. Indeed, LAG3 and CD49b are not "true" markers for human IL-10⁺ Tr1-like cells as recently reported², and we have included a discussion for this in page 13, line 273. We thank the reviewer for bringing this up for discussion.

6. Comment: "Fig 7. Expression levels of Blimp-1 after treatment with the different vectors need to be shown as per IRF4 in Supp Fig 3a."

Response: We thank the reviewer for this suggestion and have included the flow data indicating the expression of Blimp-1 in cells transduced by the related retroviral vectors, shown in the new **Fig. 8c & d**.

7. Comment: "Supp Fig 3. The addition of symbols for significant difference from naïve cells is not particularly useful and makes the figure more cluttered, could state in legend that all culture conditions had significantly increase 'xx' compared to naïve T cells. Similarly, it is not particularly informative to show differences between grouped conditions. It is important to see differences between WT and ITK^{-/-} in both control-RV and IRF4-RV groups (as already shown) as well as between WT and ITK^{-/-} cells from each group."

Response: Indeed, it is very clear that the Tr1-differentiating condition induces significantly higher level of expression of IRF4 and AHR, compared to those of the naïve condition. We have removed these symbols as suggested by the reviewer.

8. Comment: "Page 5, line 107 should refer to figure 2 not figure 1."

Response: We thank the reviewer for catching this. We have corrected this error.

9. Comment: "Page 12 lines 246-248 the sentence 'Our data suggest that re-expression of IRF4 also rescues the function of the Tr1 cells that developed suggest that this factor is sufficient, to complement ITK signals for the differentiation and function of Tr1 cells' does not make sense. Perhaps it should read more along the lines of 'the data suggest that IRF4 is sufficient to compensate for a lack of ITK as re-expression of this IRF4 rescues 'Tr1 cell' differentiation and function.'"

Response: We thank the reviewer for this suggestion, and have revised the language as suggested (revised page 12, line 240).

10. Comment: "Page 14, line 302 the phrase 'may be a promising strategy in unleashing the immune depression' does not make sense."

Response: We have changed this statement to "...may be a promising strategy in modulating immune suppression" (revised page 14, line 301).

11. Comment: "In methods section on Human Tr1 differentiation in vitro page 17 line 375 please describe what 'full RPMI-1640 medium' contains."

Response: This has been described on the revised page 19, line 381.

12. Comment: "Supp Fig 6 panel f is referred to as (b)."

Response: We thank the reviewer for catching this error, and we have corrected this.

13. Comment: “All figure legends now state an ‘n’, although some ambiguity is still present. As one example Fig 2 states n=6. Data represent results from more than three independent experiments. So data in all panels of this figure are pooled from n=6 mice from >3 experiments? Need to be more specific and state ‘data are from between 3-x independent experiments’. This should be fixed throughout all legends.”

Response: We have revised these statements to specify whether the “n” indicated are pooled numbers from several experiments or the number of replicates of a representative experiment.

We thank the reviewer for the detailed comments and helpful suggestions, and hope that these changes bring the manuscript to the level where it can be accepted for publication.

References:

1. Gagliani, N. *et al.* Coexpression of CD49b and LAG-3 identifies human and mouse T regulatory type 1 cells. *Nat Med* **19**, 739-746 (2013).
2. White, A.M. & Wraith, D.C. Tr1-Like T Cells - An Enigmatic Regulatory T Cell Lineage. *Front Immunol* **7**, 355 (2016).